# Disentangled Wasserstein Autoencoder for T-Cell Receptor Engineering

**Tianxiao Li**[1,2*], **Hongyu Guo**[3], **Filippo Grazioli**[4], **Mark Gerstein**[2†], **Martin Renqiang Min**[1†*]
[1] NEC Laboratories America, [2] Yale University
[3] National Research Council Canada, [4] NEC Laboratories Europe
tianxiao.li@yale.edu, hongyu.guo@uottawa.ca, flpgrz@outlook.com
mark.gerstein@yale.edu, renqiang@nec-labs.com

## Abstract

In protein biophysics, the separation between the functionally important residues (forming the active site or binding surface) and those that create the overall structure (the fold) is a well-established and fundamental concept. Identifying and modifying those functional sites is critical for protein engineering but computationally non-trivial, and requires significant domain knowledge. To automate this process from a data-driven perspective, we propose a disentangled Wasserstein autoencoder with an auxiliary classifier, which isolates the function-related patterns from the rest with theoretical guarantees. This enables one-pass protein sequence editing and improves the understanding of the resulting sequences and editing actions involved. To demonstrate its effectiveness, we apply it to T-cell receptors (TCRs), a well-studied structure-function case. We show that our method can be used to alter the function of TCRs without changing the structural backbone, outperforming several competing methods in generation quality and efficiency, and requiring only 10% of the running time needed by baseline models. To our knowledge, this is the first approach that utilizes disentangled representations for TCR engineering.

## 1 Introduction

Decades of work in protein biology have shown the separation of the overall structure and the smaller "functional" site, such as the generic structure versus the active site in enzymes [1], and the characteristic immunoglobulin fold versus the antigen-binding complementarity-determining region (CDR) in immunoproteins [2, 3]. The latter usually defines the protein's key function, but cannot work on its own without the stabilizing effect of the former. This dichotomy is similar to the content-style separation in computer vision [4] and natural language processing [5]. For efficient protein engineering, it is often desired that the overall structure is preserved while only the functionally relevant sites are modified. Traditional methods for this task require significant domain knowledge and are usually limited to specific scenarios. Several recent studies make use of deep generative models [6] or reinforcement learning [7] to learn from large-scale data the *implicit* generation and editing policies to alter proteins. Here, we tackle the problem by utilizing *explicit* functional features through the disentangled representation learning (DRL), where the protein sequence is separately embedded into a "functional" embedding and a "structural" embedding. This approach results in a more interpretable latent space and enables more efficient conditional generation and property manipulation for protein engineering.

DRL has been applied to the separation of "style" and "content" of images [8], or static and dynamic parts of videos [9, 10] for tasks such as style transfer [11, 8] and conditional generation [12, 13].

---

*Equal contribution. †Corresponding author.

37th Conference on Neural Information Processing Systems (NeurIPS 2023).

Attaining the aforementioned disentangled embeddings in *discrete sequences* such as protein sequences, however, is challenging because the functional residues can vary greatly across different proteins. To this end, several recent works on discrete sequences such as natural languages use adversarial objectives to achieve disentangled embeddings [14, 15]. Cheng, et al. [16] improves the disentanglement with a mutual information (MI) upper bound on the embedding space of a variational autoencoder (VAE). However, this approach relies on a complicated implementation of multiple losses that are approximated through various neural networks, and requires finding a dedicated trade-off among them, making the model difficult to train. To address these challenges, we propose a Wasserstein autoencoder (WAE) [17] framework that achieves disentangled embeddings with a theoretical guarantee, using a simpler loss function. Also, WAE could be trained deterministically, avoiding several practical challenges of VAE in general, especially on sequences [18, 19]. Our approach is proven to simultaneously maximize the mutual information (MI) between the data and the latent embedding space while minimizing the MI between the different parts of the embeddings, through minimizing the Wasserstein loss.

To demonstrate the effectiveness and utility of our method, we apply it to the engineering of T cell receptors (TCRs), which uses a similar structure fold as the immunoglobulin, one of the best-studied protein structures and a good example of separation of structure and functions. TCRs play an important role in the adaptive immune response [20] by specifically binding to peptide antigens [21] (Fig. 1A). Designing TCRs with higher affinity to the target peptide is thus of high interest in immunotherapy [22]. Various data-driven methods have been proposed to enhance the accuracy of TCR binding prediction [23–25, 12, 13, 26, 27]. However, there has been limited research on leveraging machine learning for TCR engineering. A related application is the motif scaffolding problem [28–32] where a structural "scaffold" is generated supporting a fixed functional "motif". Here our goal is the opposite: to modify the functional parts of the sequence instead by directly introducing mutations to it.

We focus on the CDR3$\beta$ region of TCRs where there is sufficient data and a clearly defined functional role (peptide binding). Using a large TCR-peptide binding dataset, we empirically demonstrate that our method successfully separates key patterns related to binding ("functional" embedding) from generic structural backbones ("structural" embedding). Due to the lack of 3D structural data, we assume that similarity between the "structure-related" parts of the sequence indicates similarity in the structure. Furthermore, by modifying only the functional embedding, our approach is able to generate new TCRs with desired binding properties while preserving the structural backbone, requiring only 10% of the running time needed by baseline models in comparison. We also note from our TCR engineering results that mutations can be introduced throughout the sequence, which implies that the model learns higher-level functional and structural patterns that could span the whole sequence, instead of looking for a universal clear cut between a "functional segment" and a "structural segment".

We summarize our main contributions as follows:

- To our knowledge, we are the first to formulate computational protein design as a style transfer problem and leverage disentangled embeddings for TCR engineering, resulting in more interpretable and efficient conditional generation and property manipulation.

- We propose a disentangled Wasserstein autoencoder with an auxiliary classifier, which effectively isolates the function-related patterns from the rest with theoretical guarantees.

- We show that by modifying only the functional embedding, we can edit TCR sequences into desired properties while maintaining their backbones, running 10 times faster than baselines.

## 2 Methods

### 2.1 Problem Formulation

We define our problem of TCR engineering task as follows: given a TCR sequence and a peptide it could not bind to, introduce a minimal number of mutations to the TCR so that it gains the ability to bind to the peptide. In the meantime, the modified TCR should remain a valid TCR, with no major changes in the structural backbone. Based on the assumption that only certain amino acids within the TCR should be responsible for peptide interactions, we can define two kinds of patterns in the TCR sequence: **functional patterns** and **structural patterns**. The former comprises the amino acids

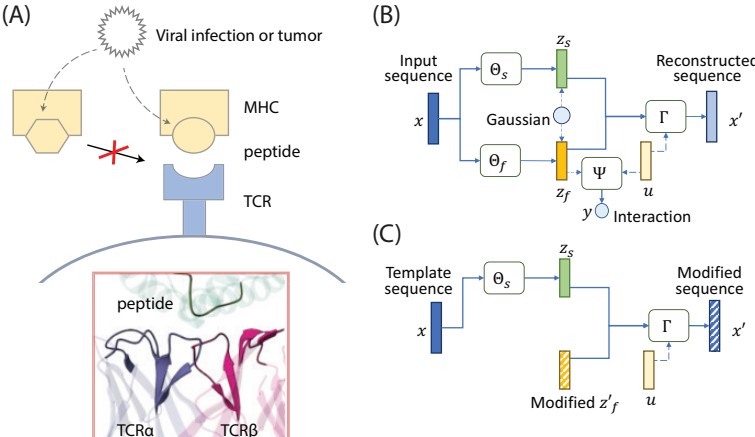

Figure 1: (A) Top: The TCR recognizes antigenic peptides provided by the major histocompatibility complex (MHC) with high specificity; bottom: the 3D structure of the TCR-peptide-MHC binding interface (PDB: 5HHO); the CDRs are highlighted. (B) The disentangled autoencoder framework, where the input $x$, i.e., the CDR3$\beta$, is embedded into a functional embedding $z_f$ (orange bar) and structural embedding $z_s$ (green bar). (C) Method for sequence engineering with input $x$. $\mathbf{z}_s$ of the template sequence and a modified $\mathbf{z}'_f$, which represents the desired peptide binding property, are fed to the decoder to generate engineered TCRs $x'$.

that define the peptide binding property. TCRs that bind to the same peptide should have similar functional patterns. The latter refers to all other patterns that do not relate to the function but could affect the validity. Following [23–25], we here limit our modeling to the CDR3$\beta$ region since it is the most active region for TCR binding. In the rest of the paper, we use "TCR" and "CDR3$\beta$" interchangeably.

## 2.2 Disentangled Wasserstein Autoencoder

Our proposed framework, named `TCR-dWAE`, leverages a disentangled Wasserstein autoencoder that learns embeddings corresponding to the functional and structural patterns. In this setting, the input data sample for the model is a triplet $\{\mathbf{x}, \mathbf{u}, y\}$, where $\mathbf{x}$ is the TCR sequence, $\mathbf{u}$ is the peptide sequence, and $y$ is the binary label indicating the interaction.

In detail, given an input triplet $\{\mathbf{x}, \mathbf{u}, y\}$, the embedding space of $\mathbf{x}$ is separated into two parts: $\mathbf{z} = \text{concat}(\mathbf{z}_f, \mathbf{z}_s)$, where $\mathbf{z}_f$ is the functional embedding, and $\mathbf{z}_s$ is the structural embedding. The schematic of the model is shown in Fig. 1B.

### 2.2.1 Encoders and Auxiliary Classifier

We use two separate encoders for the embeddings, respectively:

$$\mathbf{z}_e = \Theta_e(\mathbf{x}),$$

where $e \in \{s, f\}$ correspond to "structure" and "function".

First, the functional embedding $\mathbf{z}_f$ is encoded by the functional encoder $\Theta_f(\mathbf{x})$. In order to make sure $\mathbf{z}_f$ carries information about binding to the given peptide $\mathbf{u}$, we introduce an auxiliary classifier $\Psi(\mathbf{z}_f, \mathbf{u})$ that takes $\mathbf{z}_f$ and the peptide $\mathbf{u}$ as input and predicts the probability of positive binding label $q_\Psi(y \mid \mathbf{z}_f, \mathbf{u})$,

$$\hat{y} = q_\Psi(Y = 1 \mid \mathbf{z}_f, \mathbf{u}) = \Psi(\mathbf{z}_f, \mathbf{u}).$$

We define the binding prediction loss as binary cross entropy:

$$\mathcal{L}_{f\_cls}(\hat{y}, y) = -y \log \hat{y} - (1 - y) \log(1 - \hat{y}).$$

Second, the structural embedding $\mathbf{z}_s$ is encoded by the structural encoder $\Theta_s(\mathbf{x})$. To enforce $\mathbf{z}_s$ to contain all the information other than the peptide binding-related patterns, we leverage a sequence reconstruction loss, as will be discussed in detail in Section 2.2.3.

### 2.2.2 Disentanglement of the Embeddings

To attain disentanglement between $z_f$ and $z_s$, we introduce a Wasserstein autoencoder regularization term in our loss function following [17], by minimizing the maximum mean discrepancy (MMD) between the distribution of the embeddings $\mathbf{Z} \sim Q_{\mathbf{Z}}$ where $\mathbf{z} = \text{concat}(\mathbf{z}_f, \mathbf{z}_s)$ and an isotropic multivariate Gaussian prior $\mathbf{Z}_0 \sim P_{\mathbf{Z}}$ where $P_{\mathbf{Z}} = \mathcal{N}(0, I_d)$:

$$\mathcal{L}_{Wass}(\mathbf{Z}) = \text{MMD}(P_{\mathbf{Z}}, Q_{\mathbf{Z}}). \tag{1}$$

The MMD is estimated as follows: given the embeddings $\{\mathbf{z}_1, \mathbf{z}_2, ..., \mathbf{z}_n\}$ of an input batch of size $n$, we randomly sample from the Gaussian prior $\{\tilde{\mathbf{z}}_1, \tilde{\mathbf{z}}_2, ..., \tilde{\mathbf{z}}_n\}$ with the same sample size. We then use the linear time unbiased estimator proposed by [33] to estimate the MMD:

$$\text{MMD}(P_{\mathbf{Z}}, Q_{\mathbf{Z}}) = \frac{1}{\lfloor n/2 \rfloor} \sum_i^{\lfloor n/2 \rfloor} h((\mathbf{z}_{2i-1}, \tilde{\mathbf{z}}_{2i-1}), (\mathbf{z}_{2i}, \tilde{\mathbf{z}}_{2i})),$$

where $h((\mathbf{z}_i, \tilde{\mathbf{z}}_i), (\mathbf{z}_j, \tilde{\mathbf{z}}_j)) = k(\mathbf{z}_i, \mathbf{z}_j) + k(\tilde{\mathbf{z}}_i, \tilde{\mathbf{z}}_j) - k(\mathbf{z}_i, \tilde{\mathbf{z}}_j) - k(\mathbf{z}_j, \tilde{\mathbf{z}}_i)$ and $k$ is the kernel function. Here we use a radial basis function (RBF) [34] with $\sigma = 1$ as the kernel.

By minimizing this loss, the joint distribution of the embeddings matches $\mathcal{N}(0, I_d)$, so that $\mathbf{z}_f$ and $\mathbf{z}_s$ are independent. Also, the Gaussian shape of the latent space facilitates generation [35, 17].

### 2.2.3 Decoder and Overall Training Objective

The decoder $\Gamma$ takes $\mathbf{z}_f, \mathbf{z}_s$ and peptide $\mathbf{u}$ as input and reconstructs the original sequence as $\mathbf{x}'$. It also acts as a regularizer to enforce the structural embedding $\mathbf{z}_s$ to contain all the information other than the peptide binding-related patterns. The reconstruction loss is the position-wise binary cross entropy between $\mathbf{x}$ and $\mathbf{x}'$ averaged across all positions of the sequence:

$$\mathbf{x}' = \Gamma(\text{concat}(\mathbf{z}_s, \mathbf{z}_f, \mathbf{u}))$$

$$\mathcal{L}_{recon}(\mathbf{x}, \mathbf{x}') = \frac{1}{l} \sum_i^l -\mathbf{x}^{(i)} \log(\mathbf{x}'^{(i)}) - (1 - \mathbf{x}^{(i)}) \log(1 - \mathbf{x}'^{(i)}),$$

where $l$ is the length of the sequence and $\mathbf{x}^{(i)}$ is the probability distribution over the amino acids at the $i$-th position.

Combining all these losses with weights $\beta_1$ and $\beta_2$, we have the final objective function, which then can be optimized through gradient descent in an end-to-end fashion:

$$\mathcal{L} = \mathcal{L}_{recon} + \beta_1 \mathcal{L}_{f\_cls} + \beta_2 \mathcal{L}_{Wass}.$$

### 2.3 Disentanglement Guarantee

To show how our method can guarantee the disentangled embeddings, we provide a novel perspective on the latent space of Wasserstein autoencoders [17] utilizing the variation of information following [16]. We introduce a measurement of disentanglement as follows:

$$D(\mathbf{Z}_f, \mathbf{Z}_s; \mathbf{X} \mid \mathbf{U}) = VI(\mathbf{Z}_s; \mathbf{X} \mid \mathbf{U}) + VI(\mathbf{Z}_f; \mathbf{X} \mid \mathbf{U}) - VI(\mathbf{Z}_f; \mathbf{Z}_s \mid \mathbf{U}),$$

where $VI$ is the variation of information, $VI(\mathbf{X}; Y) = H(\mathbf{X}) + H(Y) - 2I(\mathbf{X}; Y)$, which is a measure of independence between two random variables. For simplicity, we omit the condition $\mathbf{U}$ (peptide) in the following parts.

This measurement reaches 0 when $\mathbf{Z}_f$ and $\mathbf{Z}_s$ are totally independent, i.e. disentangled. It could further be simplified as:

$$VI(\mathbf{Z}_s; \mathbf{X}) + VI(\mathbf{Z}_f; \mathbf{X}) - VI(\mathbf{Z}_f; \mathbf{Z}_s)$$
$$= 2H(\mathbf{X}) + 2[I(\mathbf{Z}_f; \mathbf{Z}_s) - I(\mathbf{X}; \mathbf{Z}_s) - I(\mathbf{X}; \mathbf{Z}_f)].$$

Note that $H(\mathbf{X})$ is a constant. Also, according to data processing inequality, as $\mathbf{z}_f \to \mathbf{x} \to y$ forms a Markov chain, we have $I(x; \mathbf{z}_f) \geq I(y; \mathbf{z}_f)$. Combining the results above, we have the upper bound of the disentanglement objective:

$$I(\mathbf{Z}_f; \mathbf{Z}_s) - I(\mathbf{X}; \mathbf{Z}_s) - I(\mathbf{X}; \mathbf{Z}_f) \leq I(\mathbf{Z}_f; \mathbf{Z}_s) - I(\mathbf{X}; \mathbf{Z}_s) - I(Y; \mathbf{Z}_f). \tag{2}$$

Next, we show how our framework could minimize each part of the upper bound in (2).

**Maximizing $I(\mathbf{X}; \mathbf{Z}_s)$** Similar to [10], we have the following theorem:

**Theorem 2.1** *Given the encoder $Q_\theta(\mathbf{Z} \mid \mathbf{X})$, decoder $P_\gamma(\mathbf{X} \mid \mathbf{Z})$, prior $P(\mathbf{Z})$, and the data distribution $P_D$*

$$\mathbb{D}_{KL}(Q(\mathbf{Z}) \parallel P(\mathbf{Z})) = \mathbb{E}_{p_D}[\mathbb{D}_{KL}(Q_\theta(\mathbf{Z} \mid \mathbf{X}) \parallel P(\mathbf{Z}))] - I(\mathbf{X}; \mathbf{Z}),$$

*where $Q(\mathbf{Z})$ is the marginal distribution of the encoder when $\mathbf{X} \sim P_D$ and $\mathbf{Z} \sim Q_\theta(\mathbf{Z} \mid \mathbf{X})$.*

Theorem 2.1 shows that by minimizing the KL divergence between the marginal $Q(\mathbf{Z})$ and the prior $P(\mathbf{Z})$, we jointly maximize the mutual information between the data $\mathbf{X}$ and the embedding $\mathbf{Z}$, and minimize the KL divergence between $Q_\theta(\mathbf{Z} \mid \mathbf{X})$ and the prior $P(\mathbf{Z})$. Detailed proof of the theorem can be found in the Appendix B.1. This also applies to the two separate parts of $\mathbf{Z}$, $\mathbf{Z}_f$ and $\mathbf{Z}_s$. In practice, because the marginal cannot be measured directly, we minimize the aforementioned kernel MMD (1) instead.

As a result, there is no need for additional constraints on the information content of $\mathbf{Z}_s$ because $I(\mathbf{X}; \mathbf{Z}_f)$ is automatically maximized by the objective. We also empirically verify in Section 3.2.3 that supervision on $\mathbf{Z}_s$ does not improve the model performance.

**Maximizing $I(Y; \mathbf{Z}_f)$** $I(Y; \mathbf{Z}_f)$ has an lower bound as follows:

$$I(Y; \mathbf{Z}_f) \geq H(Y) + \mathbb{E}_{p(Y, \mathbf{Z}_f)} \log q_\Psi(Y \mid \mathbf{Z}_f),$$

where $q_\Psi(Y \mid \mathbf{Z}_f)$ is the predicted probability by the auxiliary classifier $\Psi$ (note we omitted the condition $\mathbf{U}$ here). Thus, maximizing the performance of classifier $\Psi$ would maximize $I(Y; \mathbf{Z}_f)$.

**Minimizing $I(\mathbf{Z}_f; \mathbf{Z}_s)$** Minimization of the Wasserstein loss (1) forces the distribution of the embedding space $\mathbf{Z}$ to approach an isotropic multivariate Gaussian prior $P_\mathbf{Z} = \mathcal{N}(0, I_d)$, where all the dimensions are independent. Thus, the dimensions of $\mathbf{Z}$ will be independent, which also minimizes the mutual information between the two parts of the embedding, $\mathbf{Z}_f$ and $\mathbf{Z}_s$.

As a conclusion, our objective can jointly minimize $I(\mathbf{Z}_f; \mathbf{Z}_s)$ and maximize $I(\mathbf{X}; \mathbf{Z}_s)$ and $I(Y; \mathbf{Z}_f)$. The former ensures independence between the embeddings, and the latter enforces them to learn separate information, achieving disentanglement.

## 3 Experiments

### 3.1 Setup

**Datasets** Interacting TCR-peptide pairs are obtained from VDJDB [36], merged with experimentally-validated negative pairs from NetTCR [26]. We then expended the negatives to 5x the size of positives by adding randomly shuffled TCR-peptide pairs and only selected the peptides that could be well-classified by ERGO[23], a state-of-the-art method for TCR-binding prediction (Appendix A.2). The selected samples are then balanced resulting in 26262 positive and 26262 negative pairs with 2918 unique TCRs and 10 peptides. We also performed the same experiments on the McPAS-TCR dataset [37] (Appendix A.1). For each peptide, we select from VDJDB an additional set of 5000 TCR sequences that do not overlap with the training set. This set is used for the TCR engineering experiment.

**Implementation Details** The `TCR-dWAE` model uses two transformer encoders [38] for $\Theta_s, \Theta_f$ and a long short-term memory (LSTM) recurrent neural network decoder [39] for $\Gamma$. The auxiliary classifier $\Psi$ is a 2- layer perceptron. Hyperparameters are selected through grid search. All results are averaged across five random seeds. See Appendix B.2 for more details.

|  | $\bar{r_v}$ | $\bar{r_b}$ | %valid | #mut/len | %positive valid ↑ |
|---|---|---|---|---|---|
| TCR-dWAE | 1.36±0.03 | 0.45±0.09 | 0.61±0.06 | 0.49±0.05 | **0.23±0.02** |
| TCR-dVAE | 1.44±0.02 | 0.24±0.02 | 0.64±0.05 | 0.4±0.02 | 0.16±0.01 |
| greedy | 0.34±0.0 | 0.79±0.0 | 0.02±0.0 | 0.34±0.0 | 0.02±0.0 |
| genetic | 0.39±0.03 | 1.0±0.0 | 0.02±0.0 | NA | 0.02±0.0 |
| naive rm | 0.35±0.0 | 0.2±0.0 | 0.03±0.0 | 0.35±0.01 | 0.0±0.0 |
| MCTS | -0.1±0.0 | 0.95±0.0 | 0.0±0.0 | NA | 0.0±0.0 |
| TCR-dWAE (null) | 1.51±0.01 | 0.05±0.03 | 0.85±0.01 | 0.45±0.07 | 0.04±0.03 |
| original | 1.59 | 0.01 | 0.92 | NA | 0.01 |

Table 1: Performance comparison. `original` is the evaluation metrics on the template TCRs without any modification. Only unique sequences are counted. Results are averaged across selected peptides (i.e., SSYRRPVGI, TTPESANL, FRDYVDRFYKTLRAEQASQE, CTPYDINQM) and then averaged across five random seeds.

## 3.2 TCR Engineering

### 3.2.1 Manipulating TCR Binding via Functional Embeddings

As shown in Fig. 1C, given any TCR sequence template $\mathbf{x}$, we combine its original $\mathbf{z}_s$ and a new functional embedding $\mathbf{z}'_f$ that is positive for the target peptide $\mathbf{u}$ which is then fed to the decoder to generate a new TCR $\mathbf{x}'$ that could potentially bind to $\mathbf{u}$ while maintaining the backbone of $\mathbf{x}$. For each generation, we obtain the $\mathbf{z}'_f$ from a random positive sample in the dataset.

### 3.2.2 Metrics and Baselines

We use the following metrics to evaluate whether the engineered sequence $\mathbf{x}'$ (1) is a valid TCR sequence and (2) binds to the given peptide, which we denote as *validity score* and *binding score*, respectively:

The *validity score* $r_v$ evaluates whether the generated TCR follows similar generic patterns as naturally observed TCRs from TCRdb, an independent and much larger dataset [40]. We train another LSTM-based autoencoder on TCRdb. If the generated sequence can be reconstructed successfully by the autoencoder and has a high likelihood in the distribution of the latent space, we consider it as a valid TCR from the same distribution as the known ones (Appendix D.2). We use the sum of the reconstruction accuracy and the likelihood as the validity score and show that this metric separates true TCRs from other protein segments and random sequences (Appendix D.3).

For the *binding score*, the engineered sequence $\mathbf{x}'$ and the peptide $\mathbf{u}$ are fed into the pre-trained ERGO classifier and binding probability $r_b = \mathrm{ERGO}(\mathbf{x}', \mathbf{u})$ is calculated.

We compare `TCR-dWAE` with the following baselines (see Appendix C for more details):

**Random mutation-based** For each iteration, a new random mutation is added to the template. The best mutated sequence (one with the highest ERGO score) is selected on either a sample-level (`greedy`), population-level (`genetic`), or without any selection (`naive rm`), then goes into the next round.

**Generation-based** This includes Monte Carlo tree search (`MCTS`), where random residues are iteratively added until the sequence reaches a maximum length. For each generation process, the leaf with the highest ERGO score is added to the results.

**TCR-dVAE** We also include a VAE baseline inspired by the objectives of `IDEL` from [16], which minimizes a MI upper bound to achieve disentanglement in the embedding space (Appendix C.3).

### 3.2.3 Main Results

We use peptides with AUROC $> 0.8$ by the classifier $\Psi$ for the engineering task (SSYRRPVGI, TTPESANL, FRDYVDRFYKTLRAEQASQE, CTPYDINQM). The results are listed in Table 1. The cutoff for "valid" is $r_v \geq 1.25$, as it can successfully separate known TCRs from other protein sequences. The cutoff for "positive" TCR is $r_b > 0.5$ as the prediction scores by ERGO is rather extreme, either 0 or 1. The vast majority of the generated valid sequences (ranging from 70% ∼

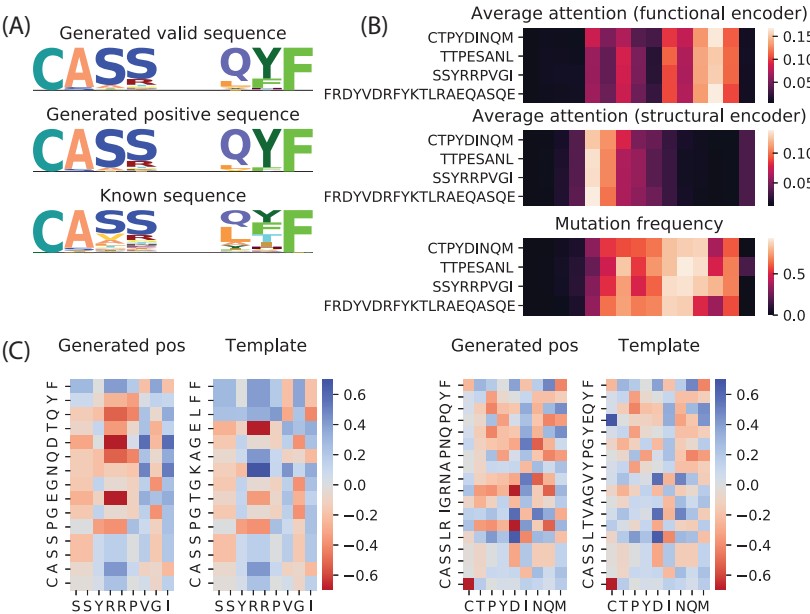

Figure 2: (A) Consensus motif of the first 4 and last 3 residues for the engineered TCRs and known TCRs; the height of the alphabet indicates the frequency of the amino acid occurring at the position. (B) (top) Average attention patterns of the functional encoder for the positive TCRs; (middle) average attention patterns of the structural encoder for the positive TCRs;(bottom) mutation frequency of the optimized TCRs compared to the templates; averaged across sequences of length 15. (C) Pairwise Miyazawa-Jernigan energy for the positive optimized sequence and its template with respect to the target peptide (left: SSYRRPVGI; right: CTPYDINQM).

100%) are unique and all are novel (Appendix Table 10). In general, `TCR-dWAE`-based methods generate more valid and positive sequences compared to other methods. We also include the results on McPAS-TCR in the Appendix Table 9, where the observations are similar.

One advantage of `TCR-dWAE` is that $z_s$ implicitly constrains the sequence backbone, ensuring validity. Methods like `genetic` and `MCTS`, on the contrary, generate much fewer valid sequences without explicit control on validity. As a result, they could produce high binding score sequences because they are guided to do so, but most of them are invalid TCRs, or out-of-distribution samples that ERGO cannot correctly predict. Also, they require calling of an external evaluation model during generation at every iteration. Adding a regularizer to enforce validity could potentially improve these baselines' generation quality, but the two objectives would be difficult to balance and will further increase the computational burden. `TCR-dWAE`, on the contrary, can perform sequence engineering in one pass, requiring 10x less time (Appendix Table 10). Compared to VAE-based models like `TCR-dVAE`, `TCR-dWAE` is a deterministic model with simpler objectives, circumventing some well-known challenges in the training of VAE which we've found to be rather sensitive to hyperparameters (see Appendix C.3 and E.5 for more comparisons between model architectures and generation modes).

### 3.2.4 Analysis of Engineered TCRs

We define the first four and the last three residues as the conservative region and the rest as the hypervariable region. On average, the sequences generated by `TCR-dAE` have a similar conservative region motif as known TCR CDR3$\beta$s [41] (Fig. 2A). The average attention score of the functional encoder concentrates on the hypervariable part (Fig. 2B, top), which has a similar distribution as the mutation frequency of the engineered sequences compared to their templates (Fig. 2B, bottom). On the other hand, we can also observe some, though less frequent, changes in the conservative regions through modifying the $z_f$, while some residues in the hypervariable region are maintained (Fig. 2C). Furthermore, the attention patterns of the functional and structural encoders, despite having their own preferences, overlap at some positions and do not have a clear-cut separation. Neither pattern

|  | %positive valid | $\mathbf{z}_f \uparrow$ | $\mathbf{z}_s \downarrow$ |
|---|---|---|---|
| full | 0.23±0.02 | 0.163±0.015 | 0.013±0.004 |
| Wass- | 0.21±0.02 | 0.040±0.026 | 0.013±0.006 |
| L_s+ | 0.18±0.01 | 0.238±0.029 | 0.008±0.000 |
| L_i- | 0.13±0.04 | 0.087±0.067 | 0.011±0.004 |

Table 2: Ablation study. Left: %positive valid for TCR engineering; Right: sample-based MMD between the embeddings of positive and negative samples. All means and standard deviations are calculated across five random seeds.

| Embedding | Average reconstruction accuracy |
|---|---|
| Original $(\mathbf{z}_f, \mathbf{z}_s)$ | 0.9044 ± 0.0020 |
| Random $\mathbf{z}_f$ | 0.7535 ± 0.0085 |
| Fully random | 0.6016 ± 0.0030 |

Table 3: Reconstruction accuracy with altered embeddings (VDJDB).

aligns with the hypervariable/conservative separation (Fig. 2B top and middle). These results indicate that both $z_f$ and $z_s$ learn to encode patterns from the entire sequence, and do not fit into the manual separation of a "functional" hypervariable region and a "structural" conservative region.

We also show some examples where residues introduced by the positive $\mathbf{z}_f$ to the engineered sequence have the potential of forming lower-energy (i.e. more stable) interactions with the target peptide (Fig. 2C), using the classic Miyazawa-Jernigan contact energy [42], which is a knowledge-based pairwise energy matrix for amino acid interactions. These results demonstrate that the positive patterns carried by $\mathbf{z}_f$ have learned some biologically relevant information such as the favored binding energy of the TCR-pMHC complex, and are successfully introduced to the engineered TCRs.

### 3.3 Ablation Study

For a quantitative comparison of the disentanglement, we calculate a sample-based MMD between the embeddings of positive and negative samples. Ideally, a disentangled latent space would make $\mathbf{z}_f$ of the positive and negative groups very different, and $\mathbf{z}_s$ indistinguishable. As shown in Table 2, removing the Wasserstein loss (Wass−) or the auxiliary classifier ($\Psi-$) leads to both poorer performance and poorer disentanglement.

We also attempt to explicitly control the information content of $\mathbf{z}_s$ by adding another decoder using only $\mathbf{z}_s$ ($L_s+$). This in practice results in similar, even slightly better, disentanglement compared to the original model, but not necessarily better performance. These observations agree with our conclusions in Section 2.3.

### 3.4 Analysis of the Embedding Space

We select a balanced subset from the test set (332 positives and 332 negatives) and obtain their $\mathbf{z}_f$ and $\mathbf{z}_s$ embeddings. T-SNE [43] visualization in Fig. 3A shows that for each peptide, the binding and non-binding TCRs can be well separated by $\mathbf{z}_f$ but not $\mathbf{z}_s$ (more examples in Appendix Fig. 5). Also, among the positive samples, $\mathbf{z}_f$ show strong clustering patterns corresponding to their peptide targets (Fig. 2B). As a result of the Wasserstein loss constraint, there is minimal correlation between $\mathbf{z}_f$ and $\mathbf{z}_s$ (Fig. 3C).

Furthermore, as shown in Table 3, the reconstruction accuracy is the highest when both $(\mathbf{z}_f, \mathbf{z}_s)$ are provided, and significantly impaired when either embedding is missing. These results imply that the embedding space is properly disentangled as intended: $\mathbf{z}_f$ and $\mathbf{z}_s$ each encodes separate parts of the sequential patterns, and both are required for faithful reconstruction.

## 4 Conclusions and Outlook

In this work, we present `TCR-dWAE`, a disentangled Wasserstein autoencoder-based framework for massive TCR engineering. The disentanglement constraints separate key patterns related to binding

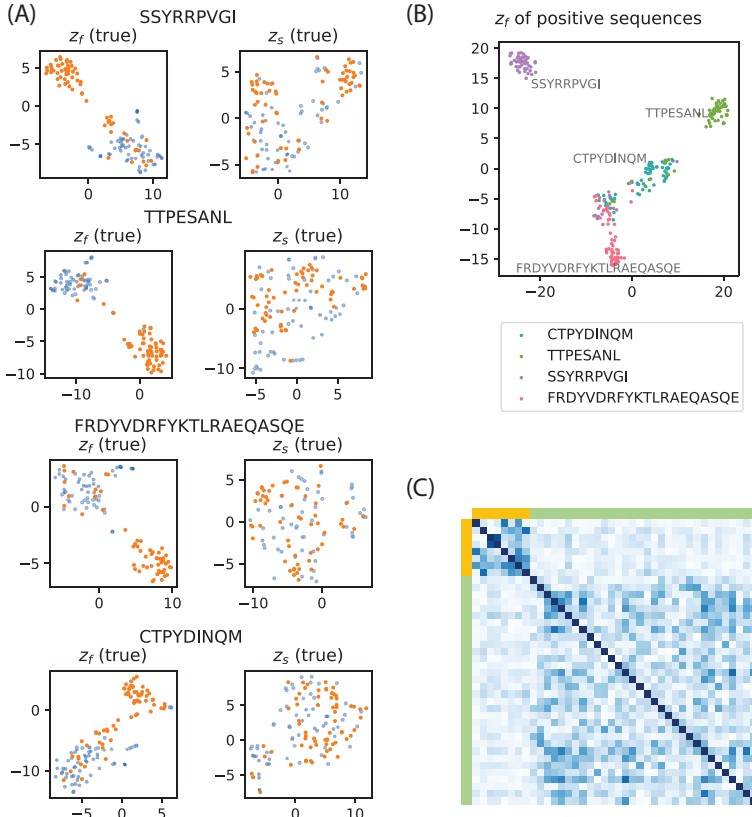

Figure 3: (A) T-SNE patterns of $\mathbf{z}_f$ and $\mathbf{z}_s$ for selected peptides, colored by ground-truth labels. (B) t-SNE of the $\mathbf{z}_f$ of the positive TCRs. Colors correspond to their binding peptide. (C) Correlation between dimensions of $\mathbf{z}_f$ and $\mathbf{z}_s$, where the orange color corresponds to $\mathbf{z}_f$ and green to $\mathbf{z}_s$.

("functional" embedding) from generic structural backbones ("structural" embedding). By modifying only the functional embedding, we are able to generate new TCRs with desired binding properties while preserving the backbone.

We are aware that our experiment is performed on a smaller subset selected by ERGO, whose limitations would bias the results. Also, our model assumes there is some "natural", yet unknown, disentanglement between features within the data. Thus, it could potentially be applied to several protein engineering tasks but would fail if the definition of functional terms is vague. With more high-throughput experimental data on TCR-peptide interaction, we will be able to train more comprehensive models on full TCR sequences. To our knowledge, ours is the first work that approaches protein sequence engineering from a style transfer perspective. As our framework mimics a vast amount of biological knowledge on function versus structure, we believe it can be further extended to a broader definition of protein functions or other molecular contexts.

## 5   Related Works

**TCR-peptide binding prediction** Most human TCRs are heterodimers comprised of a TCR$\alpha$ chain and a TCR$\beta$ chain [21]. Each chain contains three complementarity-determining regions (CDRs), CDR1, 2 and 3, which are highly variable due to gene recombination. A large amount of literature investigates the prediction of the highly specific interaction between TCRs and peptides. Early methods use consensus motifs or simple structural patterns like k-mers as predictive features [44–46, 24]. Other methods assess the binding energy for the TCR-peptide pair using physical or structural modeling [47, 48]. In recent years, several deep neural network architectures have been applied to this task [49, 26, 50, 23, 24, 51, 27], using both TCR and peptide sequences as input. Most studies

focus on the CDR3$\beta$ sequence and some also include other information like the CDR3$\alpha$, depending on the availability of the data.

**Computational biological sequence design** Traditional methods of computational biological sequence design uses search algorithms, where random mutations are introduced and then selected by specified criteria [52–54]. Deep generative models like variational autoencoder (VAE) [55] or generative adversarial network (GAN) [6] have also been successfully applied to biological sequences such as DNA. Other recent approaches include reinforcement learning (RL) [7], and iterative refinement [56]. It is also possible to co-design sequences along with the 3D structure, such as in motif scaffolding problems [28–32].

**Disentangled representation learning** The formulation of DRL tasks depends on its nature, such as separating the "style" and the "content" of a piece of text [16] or a picture [8], or static (time-invariant) and dynamic (time-variant) elements of a video [9, 10]. The disentangled representations can be used for style transfer [11, 8, 16] and conditional generation [12, 13]. Several regularization methods have been used to achieve disentanglement with or without *a priori* knowledge, including controls on the latent space capacity [57, 58], adversarial loss [12], cyclic reconstruction [11], mutual information constraints [59, 16, 9], and self-supervising auxiliary tasks [9].

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

# APPENDIX

## A  Data preparation

### A.1  Combination of data sources

TCR-peptide interaction data are obtained from VDJDB [36] and MCPAS [37]. Only peptides with $> 10$ observed pairs are used for downstream filtering. Because VDJDB and MCPAS only report interacting pairs, we first combine the dataset with the training set from NETTCR [26] which contains experimentally validated non-interacting pairs. Conflicting records are removed.

### A.2  Filtering by ERGO performance

Since ERGO [23] trains two separate models for VDJDB and MCPAS, the following filtering process is also performed separately on the two datasets. For this and all subsequent ERGO-based predictions, we use the pre-trained weights from `https://github.com/louzounlab/ERGO`.

Additional negative samples are generated as follows: a random TCR sequence is first selected from the dataset and is paired with all existing peptides in the dataset. Any unobserved pair is treated as negative. We repeat this process until the size of the negative set is 5x that of the positive set. The expanded dataset is then provided to the respective ERGO model. Predictive performance is evaluated for each peptide. We keep the peptides with AUROC and AUPR $> 0.9$ and select those among top 10 positive sample counts (Table 5).

To ensure the specificity of TCR recognition in the following study, we did a second round of filtering of both the TCRs and the peptides. We pair all TCRs with at least one positive binding event and all peptides in the filtered dataset. Any unobserved pair is treated as negative. This dataset is then provided to ERGO. Performance is shown in Table 5. We discard peptides with AUPR $< 0.7$ and TCRs that have more than one positive prediction or have at least one wrong prediction.

After that, we downsample all peptides to at most 400 positive TCRs. This number is chosen so that the resulting dataset is more balanced across peptides. The final number of samples for each peptide can be found in Table 6. To make sure the model captures peptide-specific information, for every TCR in the positive set, we add its unobserved pairings with other peptides to the negative set. We then split the TCRs into train/test/validation sets with a ratio of 8:1:1, and put all pairings of each TCR to the respective subset, to ensure all TCRs in the test and validation sets are not seen in the training. For the training set, the positive samples are up-sampled by the negative/positive ratio of the original dataset.

## B  Model details

| Notation | Meaning |
|---|---|
| $\Theta_f$ | functional encoder |
| $\Theta_s$ | structural encoder |
| $\Gamma$ | decoder |
| $\Psi$ | auxiliary functional classifier |
| $\{\mathbf{x}, \mathbf{u}, y\}$ | a data point with TCR $\mathbf{x}$, peptide $\mathbf{u}$ and binding label $y$ |
| $\mathbf{z}_f$ | functional embedding |
| $\mathbf{z}_s$ | structural embedding |
| $\mathbf{z}$ | concatenation of $\{\mathbf{z}_f, \mathbf{z}_s\}$ |
| $\mathbf{x}'$ | reconstructed/generated sequence from the decoder |
| $\mathbf{x}^{(i)}$ | the probability distribution over amino acids at the $i$-th position in $\mathbf{x}$ |
| $\mathrm{concat}(\mathbf{x}_1, ..., \mathbf{x}_n)$ | concatenation of vectors $\{\mathbf{x}_1, ..., \mathbf{x}_n\}$ |

Table 4: Notations used for this paper. Sequences are represented as $l \times |V|$ matrices where $l$ is the length $|V|$ is the number of amino acids.

### B.1  Proof of Theorem 1

We use density functions for simplicity. Let $q_\theta(\mathbf{z} \mid \mathbf{x})$ be the encoder and $p_\gamma(\mathbf{x} \mid \mathbf{z})$ be the decoder. We have the joint generative distribution:

$$p(\mathbf{x}, \mathbf{z}) = p_\gamma(\mathbf{x} \mid \mathbf{z})p(\mathbf{z}),$$

where $p(\mathbf{z})$ is the prior. Also, we have the joint inference distribution:
$$q(\mathbf{x}, \mathbf{z}) = q_\theta(\mathbf{z} \mid \mathbf{x}) p_D(\mathbf{x}),$$
where $p_D(\mathbf{x})$ is the data distribution.

$$
\begin{aligned}
I(\mathbf{X}; \mathbf{Z}) &= \mathbb{E}_{q(\mathbf{x}, \mathbf{z})} \log \frac{q(\mathbf{x}, \mathbf{z})}{p_D(\mathbf{x}) q(\mathbf{z})} \\
&= \mathbb{E}_{p_D} \sum_{\mathbf{z}} p_D(\mathbf{x}) q_\theta(\mathbf{z} \mid \mathbf{x}) \log \frac{q_\theta(\mathbf{z} \mid \mathbf{x}) p_D(\mathbf{x})}{p_D(\mathbf{x}) q(\mathbf{z})} \\
&= \mathbb{E}_{p_D} \sum_{\mathbf{z}} q_\theta(\mathbf{z} \mid \mathbf{x}) \log \frac{q_\theta(\mathbf{z} \mid \mathbf{x})}{q(\mathbf{z})} \\
&= \mathbb{E}_{p_D} \sum_{\mathbf{z}} q_\theta(\mathbf{z} \mid \mathbf{x}) \log \frac{q_\theta(\mathbf{z} \mid \mathbf{x})}{p(\mathbf{z})} - \mathbb{E}_{p_D} \sum_{\mathbf{z}} q_\theta(\mathbf{z} \mid \mathbf{x}) \log \frac{q(\mathbf{z})}{p(\mathbf{z})} \\
&= \mathbb{E}_{p_D} \sum_{\mathbf{z}} q_\theta(\mathbf{z} \mid \mathbf{x}) \log \frac{q_\theta(\mathbf{z} \mid \mathbf{x})}{p(\mathbf{z})} - \sum_{\mathbf{z}} q(\mathbf{z}) \log \frac{q(\mathbf{z})}{p(\mathbf{z})} \\
&= \mathbb{E}_{p_D} [\mathbb{D}_{\mathrm{KL}}(Q_\theta(\mathbf{Z} \mid \mathbf{X}) \parallel P(\mathbf{Z}))] - \mathbb{D}_{\mathrm{KL}}(Q(\mathbf{Z}) \parallel P(\mathbf{Z})).
\end{aligned}
$$
Thus,
$$\mathbb{D}_{\mathrm{KL}}(Q(\mathbf{Z}) \parallel P(\mathbf{Z})) = \mathbb{E}_{p_D} [\mathbb{D}_{\mathrm{KL}}(Q_\theta(\mathbf{Z} \mid \mathbf{X}) \parallel P(\mathbf{Z}))] - I(\mathbf{X}; \mathbf{Z}).$$

## B.2 Implementation and training details

All input sequences are padded to the same length (25). The peptide $\mathbf{u}$ is represented as the average BLOSUM50 score [60] for all its amino acids. The model is trained from end to end using the Adam optimizer [61]. The first layer of the model is an embedding layer that transforms the one-hot encoded sequence $\mathbf{x}$ into continuous-valued vectors of 128 dimensions:
$$\mathbf{e} = W^{emb} \mathbf{x}.$$
Both $\mathbf{z}_f$ and $\mathbf{z}_s$ encoders are 1-layer transformer encoders with 8 attention heads and an intermediate size of 128. The transformer layer utilizes the multi-head self-attention mechanism. For each attention head $i$:
$$Q_i = W_i^Q \mathbf{e}, K_i = W_i^K \mathbf{e}, V_i = W_i^V \mathbf{e}$$
$$\mathrm{Attn}_i(\mathbf{e}) = \mathrm{softmax}\left(\frac{Q_i K_i^T}{\sqrt{d_k}}\right) V_i,$$
where $d_k$ is the dimension of $Q_i$ and $K_i$. The outputs of the attention heads are then aggregated as follows:
$$\mathrm{Multihead}(\mathbf{e}) = \mathrm{concat}(\mathrm{Attn}_1(\mathbf{e}), \mathrm{Attn}_2(\mathbf{e}), ...) W^O.$$
A 2-layer MLP with a 128-dimension hidden layer is then built on top of the transformer (which has the same dimension as the input embeddings) to transform the output to the dimensions of $\mathbf{z}_f$ and $\mathbf{z}_s$, respectively. The functional classifier is a 2-layer MLP with a 32-dimension hidden layer. The decoder is a 2-layer LSTM with 256 hidden dimensions.

The hyperparameters are selected with grid search and models with the best generation results are reported. Specifically, weights of all losses are selected from $[1.0, 0.1]$. The dimension of $\mathbf{z}_f$ is fixed to 8 and $\mathbf{z}_s$ to 32. We train each model with 200 epochs and a learning rate of $1e - 4$, and evaluate the checkpoint of every 50 epochs. We find the variance of the RBF kernel (for the calculation of the Wasserstein loss) does not have a strong impact on the results significantly, so the value is fixed to 1.0.

The model is trained with five different random seeds (42, 123, 456, 789, 987). We report the hyperparameter setting with the best average performance (i.e. one that generates the highest average number of qualified positive sequences for the well-classified peptides).

The hyperparameter settings of the models for comparison and visualization are:
$$[\beta_1 = 1.0, \beta_2 = 0.1, \mathrm{epoch} = 200]$$
where $\beta$'s are weights of the losses:
$$\mathcal{L} = \mathcal{L}_{recon} + \beta_1 \mathcal{L}_{f\_cls} + \beta_2 \mathcal{L}_{Wass}.$$
For the visualization and analysis of the model trained on VDJDB, we use random seed $= 789$.

We use the scheduled sampling technique [62] for the LSTM decoder during training, where for each position in the input sequence, there is a $0.5$ probability of using the previous predicted token, instead of the original token, to calculate the hidden state for the next position. This is employed to avoid the discrepancy between the training and the generation, as the former uses the original sequence to calculate the hidden states and the latter uses predicted tokens.

The model is trained on 2 rtx3090 GPUs with a batch size of 256 (128 per GPU). Training with 200 epochs typically takes $\sim 4$ hours.

## C  Baseline methods

We compare our model with two types of methods for the generation of the optimized TCR $\mathbf{x}'$: (1) mutation-based, which iteratively adds random mutations to the template sequence; and (2) generation-based, which generates novel sequences of the pre-determined length range. For both types of methods, the modified/generated sequences are selected by peptide binding scores from the respective pre-trained ERGO. The experiments are performed on each peptide in the dataset independently.

### C.1  Mutation-based baselines

**Random mutation**  (`naive rm`) The TCR is randomly mutated by one amino acid for 8 times progressively. This process is repeated for 10 runs for each TCR and the resulting one with the highest ERGO prediction score is reported.

**Greedy mutation**  (`greedy`) For each TCR, 10 randomly mutated sequences are generated, each with one amino acid difference from the original sequence. Among the 10 mutated sequences, we select the one that gives the highest binding prediction with the given peptide as the template for the next run. This process is repeated 8 times.

**Genetic algorithm**  (`genetic`) Let $M$ be the sample size. For each TCR, 10 randomly mutated sequences are generated, each with one amino acid difference from the original sequence. All mutated sequences along with the original TCRs are then pooled together, and the top $M$ sequences that give the highest binding prediction are used as the input for the next run. This process is repeated 8 times.

### C.2  Generation-based baselines

**Monte Carlo tree search**  (`MCTS`) TCRs are generated by adding amino acids iteratively, resulting in a search tree. When the TCR length reaches 10, the binding score is estimated by ERGO. For each iteration, a random node is selected for expansion and then evaluated by ERGO, and the scores of all its parent nodes are updated accordingly. The tree expansion ends when the length reaches 20. For every generation process, the highest leaf node is added to the output TCR set.

### C.3  TCR-dVAE

TCR-dVAE uses a similar objective as IDEL [16], which is a VAE with a mutual information constraint on the latent space. For training, the loss comprises of the following components:

- The reconstruction loss: $\mathcal{L}_{recon}(\mathbf{x}, \mathbf{x}')$
- The KL divergence term for VAE: $\mathcal{L}_{KL} = \mathbb{D}_{\mathrm{KL}}(q_\theta(\mathbf{z}_s, \mathbf{z}_f|\mathbf{x}) \parallel p(\mathbf{z}_s, \mathbf{z}_f))$.
- The reconstruction loss given $\mathbf{z}_s$: $\mathcal{L}_s(\mathbf{x}, \Phi(\mathbf{z}_s))$ where $\Phi$ is an auxiliary decoder.
- The classification loss given $z_f$: $\mathcal{L}_{f\_cls}(\hat{y}, y)$
- The sample-based MI upper bound between the embeddings: $\mathcal{L}_{MI}(\mathbf{z}_f, \mathbf{z}_s)$. This requires an approximation of the conditional distribution $p(\mathbf{z}_f|\mathbf{z}_s)$, which is achieved by a separate neural network.

Here we use our own notations, not the ones used in the original paper, for better comparison.

For the hyperparameters, we use 10.0 weight for $\mathcal{L}_{recon}$, $\mathcal{L}_s$ and $\mathcal{L}_{f\_cls}$, 1.0 for the other terms, and a learning rate of $1e-4$. In practice, we perform annealing [18] on $\mathcal{L}_{KL}$ and $\mathcal{L}_{MI}$ where their weights gradually increase from 0 to 1 during the first 100 epochs and remain for the rest of the epochs, to make sure the embeddings are as informative as possible. We observe that other hyperparameter settings would easily lead to severe posterior collapse where the embedding is ignored for the reconstruction.

## D  Evaluation of the optimized sequences

### D.1  Training of the autoencoder

We train an LSTM-based autoencoder, which we denote as TCR-AE0, on the 277 million TCR sequences from TCRdb [40]. TCR-AE0 has a latent space of dimension 16 and is trained for 50,000 steps with a batch size of 256.

## D.2 Validity score

The validity score combines two scores calculated from TCR-AE0:

- The reconstruction-based score is calculated as
$$r_r(\mathbf{x}') = 1 - \text{lev}(\mathbf{x}', \text{TCR-AE0}(\mathbf{x}'))/l(\mathbf{x}'),$$
where $\text{lev}(\mathbf{x}', \text{TCR-AE0}(\mathbf{x}'))$ is the Levenstein distance between the original sequence and the reconstructed sequence, and $l(\mathbf{x}')$ is the length of the reconstructed sequence. Higher $r_r$ means $\mathbf{x}'$ is better reconstructed from TCR-AE0 and is thus more likely to be a valid TCR sequence.

- The density-based score calculates whether the embedding of $\mathbf{x}'$ follows the same distribution as known TCRs. We learn a Gaussian mixture model from the latent embeddings of known TCRs from TCR-AE0. The likelihood of the embedding $\mathbf{e}'$ of $\mathbf{x}'$ from TCR-AE0 falling in the same Gaussian mixture distribution is denoted as $P(\mathbf{e}')$. The density-based score is calculated as
$$r_d(\mathbf{x}') = \exp(1 + \frac{\log P(\mathbf{e}')}{\tau}),$$
where $\tau = 10$. Higher $r_d$ means the latent embedding of $\mathbf{x}'$ from TCR-AE0 is more likely to follow the same distribution as other valid TCR sequences.

We then define the validity score as $r_v = r_r + r_d$.

## D.3 Validation of the metrics

We compare the TCR-AE-derived evaluation metric scores of three different sources:

(1) all unique CDR3$\beta$ sequences from VDJDB.

(2) random segments of length $8 - 18$ (which is the most frequent lengths of CDR3$\beta$ sequences) from random uniport [63] protein sequences of the same size as (1). The conservative 'C' at the beginning and 'F' at the end are added to the segments.

(3) random shuffling of the sequences from (1), where the first 'C' and the last 'F' are kept

We show in Fig. 4 that for both two scores $r_d$ and $r_r$, as well as their sum, CDR3$\beta$ sequences score much higher than random proteins or shuffled sequences. This shows these scores could be effectively used for the estimation of TCR sequence validity. We choose $r_v > 1.25$ as the criteria for valid sequences as it rejects most negatives.

# E Extended Results

## E.1 Comparison of TCR Engineering Performance

We find consistently improved performance of our method over the baselines in both VDJDB (Table 1) and McPAS-TCR (Table 9). Also, the majority of generated sequences are unique (Table 10) and all are novel (not observed in the original dataset; statistics not shown).

The relative performance between methods in comparison still holds when all peptides, including those that are poorly classified, are taken into account (Table 7). However, when the results are separated by individual peptides (Table 8), we observe that for the poorly classified peptides (AUC<0.8; see Methods), using a positive $\mathbf{z}_f$ (`TCR-dWAE` row) has little difference from using a random $\mathbf{z}_f$ (`TCR-dWAE (null)` row). This further highlights the importance of well-defined and well-classified functional labels. Otherwise, $\mathbf{z}_f$ does not encode the proper functional information, so it becomes uninformative and does not alter the sequence function better than a random embedding.

## E.2 Analysis of the Model

We show extended $\mathbf{z}_f$ and $\mathbf{z}_s$ T-SNE patterns in Fig. 5, colored by the ground truth label as well as the predicted label. For the well-classified peptides, there is a clear separation of positives and negatives in the $\mathbf{z}_f$ space but not $\mathbf{z}_s$. There are cases where the true positives are not separable from the true negatives using $\mathbf{z}_f$, but the predicted positives and the predicted negatives (by the function classifier $\Psi$) are still separated. We consider the latter as a problem with data quality and classification accuracy, not embedding. Meanwhile, the classifier shows consistent performance over the peptides across random seeds with (Fig. 6, left) and without (Fig. 6, right) the Wasserstein loss.

As a result of the Wasserstein loss, the distribution of the embedding space is closer to a multivariate Gaussian (Fig. 7A. It becomes less regularized without the Wasserstein loss (Fig. 7B). Contrary to $\mathbf{z}_f$ (Fig. 3B in the main text), the T-SNE of $\mathbf{z}_s$ and first-layer embedding of the encoder for the positive samples cannot distinguish the binding targets from each other (Fig. 8A).

## E.3 Analysis of the Generated Sequences

In addition to the results presented in the main text, we also selected 500 random positive and negative sequences from the training set and replaced their $\mathbf{z}_f$ with the most positive/negative one in the subset. The generated sequences using their original $\mathbf{z}_s$ and the new $\mathbf{z}_f$ have binding scores mostly related to the $\mathbf{z}_f$, regardless of whether the $\mathbf{z}_s$ source is positive or negative. This shows $\mathbf{z}_f$ can be used to encode and transfer binding information, which lays the foundation for the following TCR engineering experiments (Fig. 8B).

The generated sequences have a similar length distribution as their templates (Fig. 8C), meaning no drastic changes are made. We further find that the $\mathbf{z}_s$ of the modified TCRs show high cosine similarity with those of their templates, while the $\mathbf{z}_f$ are more similar to the $\mathbf{z}_f$ used for their generation (Fig. 8D), but not with that of the template. These show that the modified TCRs preserve the "structural" information from $\mathbf{z}_s$ and incorporate the new "functional" information from the modified $\mathbf{z}_f$.

As in Fig. 8E, there is no significant correlation between the sequence validity and the binding scores. Also, we find that most binding scores predicted by ERGO are rather extreme (either 1 or 0).

## E.4 Interpolation

For each peptide, we perform interpolation of $\mathbf{z}_f$ between 100 random pairs for 10 steps with $\mathbf{z}_f^{(r)} = r\mathbf{z}_f^{(1)} + (1-r)\mathbf{z}_f^{(2)}, r \in [0, 1]$, while $\mathbf{z}_s$ remains the same. Fig. 8F shows that interpolating between positive $\mathbf{z}_f$ pairs preserves the positive binding score compared to negative pairs. Here we show one example of interpolation, where the first column is the generated sequence and the second column is the predicted binding score:

Seq 1: `CASTESDRRSQNTQYF`
Seq 2: `CASSLSTFTANTAQLFF`
Peptide: `CTPYDINQM`

Interpolated $\mathbf{z}_f$, with $\mathbf{z}_s$ of Seq 1

| Sequence | $r_b$ |
| --- | --- |
| `CASTESDRRSQNTQYF` | 1.0 (original) |
| `CASTESDRRSQNTQYF` | 1.0 |
| `CASTESDRRSQNTQYF` | 1.0 |
| `CASTESDRNSNQPQYF` | 1.0 |
| `CASTESDRNSNQPQYF` | 1.0 |
| `CASTESTKNSNQPQYF` | 1.0 |
| `CASTESTKNSNQPQYF` | 1.0 |
| `CASSESTMNSNQPQYF` | 1.0 |
| `CASSESTTNSNQPQYF` | 1.0 |
| `CASSESTTNSNQPQYF` | 1.0 |
| `CASSESTTANTAQLFF` | 1.0 |

Interpolated $\mathbf{z}_f$, with $\mathbf{z}_s$ of Seq 2

| Sequence | $r_b$ |
| --- | --- |
| `CASSLSTFTANTAQLFF` | 1.0 (original) |
| `CASSLSTFTANTAQLFF` | 1.0 |
| `CASSLSTFTANTAQLFF` | 1.0 |
| `CASSLSTFTANTAQLFF` | 1.0 |
| `CASSLSTFTANTAQLFF` | 1.0 |
| `CASSLSTFTANTAQLFF` | 1.0 |
| `CASSLSTRTANTAQLFF` | 1.0 |
| `CASSLSTRTANNTQLFF` | 0.16 |
| `CASSLSTRTAKNTQLFF` | 0.97 |
| `CASSLSTRTSQNTQYF` | 1.0 |
| `CASSLSTRDSTNTQYF` | 1.0 |

These results further prove that $\mathbf{z}_s$ could be used to transfer binding information independent of the $\mathbf{z}_s$, and that the latent space is well-regularized. Also, throughout the interpolation, the hypervariable region only undergoes minor changes. This shows $\mathbf{z}_s$ could preserve the structural backbone information within both the conservative and hypervariable regions.

### E.5 Extended comparison between sampling methods and model architecture

We experiment with several different generation modes and model architecture. The results are shown in Table 11.

### E.5.1 Source of positive embeddings

In addition to using a $\mathbf{z}_f$ from a random positive sample (labeled as `random pos`), we also try two other means of obtaining positive $\mathbf{z}_f$'s: (1) `best`: we use the $\mathbf{z}_f$ of the sample with the highest classification score from the auxiliary classifier $\Psi$ (2) `avg`: we use the average $\mathbf{z}_f$ for all positive samples. In both cases, the same $\mathbf{z}_f$ is used for all generations. We perform the same experiments with `TCR-dVAE`, by using the mean of the $\mathbf{z}_f$ of the positive sample. We observe that all three methods have comparable performance for `TCR-dWAE`, indicating a well-regularized latent space.

### E.5.2 Stochastic generation

As `TCR-dVAE` is a probabilistic model, we also compared with a random sampling-based method. We use both the predicted mean and variance of the positive source to generate 5 random $\mathbf{z}_f$'s and take the average results of all the generated samples (labeled `TCR-dVAE-rand`). In this scenario, only the `random pos` and `best` modes are applicable. We observe a notable decrease in the performance as the quality of generation varies with the random sampling, probably due to a less regularized latent space.

### E.5.3 Transformer decoder

We further test a `TCR-dWAE` model with a transformer decoder (`TCR-dWAE-attn`) instead of a LSTM decoder. The performance is comparable with those with LSTM (Table 11) while the autoregressive generation is notably slower (Table 10)

### E.5.4 Modeling the hypervariable region only

We train a model only using the hypervariable region and run the sequence modification pipeline as in full CDR3beta (`TCR-dWAE (trimmed)`). After the hypervariable region is modified, we add back the conservative region of the input template for each generated sequence.

As shown in Table 11, the performance is worse than modeling the full sequence. To our surprise, it seems that the lower performance is caused by a massive drop in the number of positive sequences, while the sequence validity is not affected much by the trimming, as can be seen from the number of valid sequences. This may indicate some interactions between the two regions that are essential for positive binding. Thus, the modeling and generation should be performed on the entire CDR3$\beta$ region, instead of on manually defined segments.

### E.6 Comparison between evaluation metrics

### E.6.1 Binding prediction

Besides ERGO [23], NetTCR[26] is another sequence-based TCR-peptide binding model with competitive performance. However, NetTCR is only trained on three peptides and shows the best performance when both the $\alpha$ and $\beta$ chain are provided, which is mostly not available in our dataset. Thus, ERGO provides a more efficient evaluation method for single-chain data and a greater variety of peptides. For the comparison between model architectures, we re-train a single-chain NetTCR on our own training set with more peptides. The evaluation result on the generated sequences in Table 1 in the manuscript (reporting the %positive valid only) with the two classifiers is as follows. We show in Table 12 that the conclusions are similar between the two models.

### E.6.2 Uniqueness and novelty score

We calculate the novelty score following [64] for each individual sequence from the positive valid subset in Table 1:
$$\text{Nov}(x) = \min_{s_i \in D_0} d(x, s_i)$$
where $D_0$ is the set of templates and the distance measure $d$ is the Levenshtein distance relative to the length of the longer sequence. This score can then be used to select the "novel" sequences. We also include a held-out validation set as a reference. We find that around $1/3 \sim 1/2$ of the sequences remain at cutoff 0.2 and all are excluded at cutoff 0.4 (Table 14). The same pattern is observed in the validation set, which indicates a similar distribution of "novelty" between the generated sequences and the raw data.

We also report the same novelty and diversity metrics for the positive valid subset as in [64]. We would like to point out that:

(1) These metrics evaluate the set of generated sequences as a whole, while we directly select the sequences that meet the validity and binding score criterion, so they are not our primary objective.

(2) The comparisons are only meaningful within different generative models. For the random mutation-based methods, one can always achieve high novelty and diversity by introducing a sufficient number of mutations.

We show that the results are not very different between TCR-dWAE and TCR-dVAE (Table 13). In addition, we include the case of a VAE that suffers from posterior collapse (where the generation is always limited to the same few patterns). The scores for the first two are at a similar level as the validation set, while the diversity score for the collapsed VAE drops a lot. These suggest that the positive valid sequences presented in the paper are sufficiently diverse and novel.

| | source | #pos | auroc | aupr |
|---|---|---|---|---|
| AVFDRKSDAK | vdjdb | 1641 | 0.94 | 0.71 |
| CTPYDINQM | vdjdb | 500 | 0.99 | 0.81 |
| ELAGIGILTV | vdjdb | 1410 | 0.95 | 0.79 |
| FRDYVDRFYKTLRAEQASQE | vdjdb | 367 | 0.98 | 0.85 |
| GILGFVFTL | vdjdb | 3408 | 0.95 | 0.89 |
| GLCTLVAML | vdjdb | 962 | 0.92 | 0.73 |
| IVTDFSVIK | vdjdb | 548 | 0.94 | 0.62 |
| KRWIILGLNK | vdjdb | 319 | 0.95 | 0.54 |
| NLVPMVATV | vdjdb | 4421 | 0.94 | 0.85 |
| RAKFKQLL | vdjdb | 830 | 0.94 | 0.75 |
| SSLENFRAYV | vdjdb | 322 | 0.99 | 0.57 |
| SSYRRPVGI | vdjdb | 337 | 0.99 | 0.81 |
| STPESANL | vdjdb | 234 | 0.99 | 0.35 |
| TTPESANL | vdjdb | 511 | 0.99 | 0.75 |
| ASNENMETM | mcpas | 265 | 0.98 | 0.63 |
| CRVLCCYVL | mcpas | 435 | 0.95 | 0.7 |
| EAAGIGILTV | mcpas | 272 | 0.97 | 0.55 |
| FRCPRRFCF | mcpas | 266 | 0.96 | 0.58 |
| GILGFVFTL | mcpas | 1142 | 0.96 | 0.9 |
| GLCTLVAML | mcpas | 828 | 0.95 | 0.85 |
| LPRRSGAAGA | mcpas | 2142 | 0.96 | 0.88 |
| NLVPMVATV | mcpas | 543 | 0.93 | 0.78 |
| RFYKTLRAEQASQ | mcpas | 304 | 0.99 | 0.91 |
| SSLENFRAYV | mcpas | 416 | 0.99 | 0.78 |
| SSYRRPVGI | mcpas | 337 | 0.99 | 0.83 |
| TPRVTGGGAM | mcpas | 274 | 0.95 | 0.52 |
| VTEHDTLLY | mcpas | 273 | 0.95 | 0.45 |
| WEDLFCDESLSSPEPPSSSE | mcpas | 364 | 0.98 | 0.93 |

Table 5: Statistics and ERGO prediction performance for the selected peptides from the first round.

| VDJDB | | | | MCPAS | | |
|---|---|---|---|---|---|---|
| | #pos | #all | | | #pos | #all |
| NLVPMVATV | 2880 | 5478 | | NLVPMVATV | 1792 | 3810 |
| GLCTLVAML | 2880 | 5478 | | RFYKTLRAEQASQ | 1528 | 3579 |
| RAKFKQLL | 2880 | 5478 | | WEDLFCDESLSSPEPPSSSE | 1928 | 3929 |
| AVFDRKSDAK | 2880 | 5478 | | GILGFVFTL | 2560 | 4482 |
| SSYRRPVGI | 2268 | 4934 | | SSYRRPVGI | 1504 | 3558 |
| GILGFVFTL | 2880 | 5478 | | SSLENFRAYV | 1824 | 3838 |
| TTPESANL | 2286 | 4950 | | CRVLCCYVL | 1680 | 3712 |
| FRDYVDRFYKTLRAEQASQE | 2034 | 4726 | | LPRRSGAAGA | 2560 | 4482 |
| ELAGIGILTV | 2880 | 5478 | | GLCTLVAML | 2560 | 4482 |
| CTPYDINQM | 2394 | 5046 | | | | |

Table 6: Statistics of the training data by peptide.

| VDJDB | | | | | |
|---|---|---|---|---|---|
| | $\bar{r_v}$ | $\bar{r_b}$ | %valid | #mut/len | %positive valid |
| TCR-dWAE | 1.47±0.02 | 0.31±0.04 | 0.76±0.03 | 0.47±0.06 | 0.18±0.01 |
| TCR-dVAE | 1.51±0.01 | 0.2±0.01 | 0.73±0.02 | 0.38±0.02 | 0.15±0.01 |
| greedy | 0.31±0.0 | 0.88±0.0 | 0.02±0.0 | 0.32±0.0 | 0.02±0.0 |
| genetic | 0.36±0.02 | 1.0±0.0 | 0.02±0.0 | 1.0±0.0 | 0.02±0.0 |
| naive rm | 0.33±0.0 | 0.39±0.0 | 0.02±0.0 | 0.35±0.0 | 0.01±0.0 |
| MCTS | -0.1±0.0 | 0.95±0.0 | 0.0±0.0 | 0.0±0.0 | 0.0±0.0 |
| TCR-dWAE (null) | 1.51±0.01 | 0.08±0.02 | 0.86±0.01 | 0.45±0.06 | 0.07±0.02 |
| original | 1.59 | 0.03 | 0.92 | 0.0 | 0.03 |

Table 7: Performance comparison for VDJDB, averaged across **all** peptides (ELAGIGILTV, GLCTL-VAML, AVFDRKSDAK, SSYRRPVGI, RAKFKQLL, CTPYDINQM, TTPESANL, NLVPMVATV, FRDYVDRFYKTLRAEQASQE, GILGFVFTL )

| | $\bar{r_v}$ | $\bar{r_b}$ | %valid | #mut/len | %positive valid |
|---|---|---|---|---|---|
| | | | SSYRRPVGI | | |
| TCR-dWAE | 1.11±0.14 | 0.48±0.18 | 0.37±0.16 | 0.48±0.06 | 0.06±0.01 |
| TCR-dVAE | 1.35±0.06 | 0.25±0.07 | 0.68±0.05 | 0.38±0.02 | 0.06±0.02 |
| TCR-dWAE (null) | 1.5±0.01 | 0.03±0.05 | 0.85±0.01 | 0.43±0.05 | 0.02±0.04 |
| original | 1.59 | 0.02 | 0.92 | 0.0 | 0.01 |
| | | | TTPESANL | | |
| TCR-dWAE | 1.41±0.06 | 0.64±0.11 | 0.56±0.06 | 0.52±0.04 | 0.34±0.06 |
| TCR-dVAE | 1.42±0.02 | 0.4±0.07 | 0.74±0.02 | 0.46±0.01 | 0.19±0.01 |
| TCR-dWAE (null) | 1.5±0.01 | 0.05±0.04 | 0.85±0.01 | 0.46±0.08 | 0.04±0.03 |
| original | 1.59 | 0.01 | 0.92 | 0.0 | 0.01 |
| | | | FRDYVDRFYKTLRAEQASQE | | |
| TCR-dWAE | 1.59±0.03 | 0.28±0.06 | 0.86±0.03 | 0.47±0.07 | 0.22±0.04 |
| TCR-dVAE | 1.58±0.02 | 0.35±0.06 | 0.91±0.01 | 0.39±0.02 | 0.22±0.01 |
| TCR-dWAE (null) | 1.51±0.01 | 0.05±0.04 | 0.86±0.01 | 0.45±0.07 | 0.04±0.03 |
| original | 1.59 | 0.02 | 0.92 | 0.0 | 0.01 |
| | | | CTPYDINQM | | |
| TCR-dWAE | 1.36±0.03 | 0.51±0.07 | 0.66±0.04 | 0.49±0.05 | 0.29±0.04 |
| TCR-dVAE | 1.41±0.02 | 0.36±0.06 | 0.76±0.03 | 0.43±0.02 | 0.18±0.02 |
| TCR-dWAE (null) | 1.5±0.01 | 0.08±0.02 | 0.85±0.01 | 0.45±0.07 | 0.06±0.02 |
| original | 1.59 | 0.02 | 0.92 | 0.0 | 0.01 |
| | | | ELAGIGILTV* | | |
| TCR-dWAE | 1.56±0.03 | 0.1±0.01 | 0.9±0.02 | 0.44±0.06 | 0.09±0.01 |
| TCR-dVAE | 1.58±0.02 | 0.13±0.02 | 0.92±0.02 | 0.32±0.02 | 0.1±0.01 |
| TCR-dWAE (null) | 1.51±0.02 | 0.08±0.0 | 0.86±0.02 | 0.45±0.05 | 0.07±0.0 |
| original | 1.59 | 0.05 | 0.92 | 0.0 | 0.04 |
| | | | GLCTLVAML* | | |
| TCR-dWAE | 1.56±0.02 | 0.11±0.01 | 0.89±0.02 | 0.47±0.06 | 0.1±0.01 |
| TCR-dVAE | 1.58±0.02 | 0.14±0.03 | 0.91±0.02 | 0.35±0.03 | 0.11±0.02 |
| TCR-dWAE (null) | 1.51±0.01 | 0.09±0.0 | 0.85±0.01 | 0.46±0.06 | 0.07±0.01 |
| original | 1.59 | 0.04 | 0.92 | 0.0 | 0.04 |
| | | | AVFDRKSDAK* | | |
| TCR-dWAE | 1.58±0.03 | 0.15±0.01 | 0.9±0.02 | 0.45±0.06 | 0.13±0.01 |
| TCR-dVAE | 1.59±0.02 | 0.19±0.02 | 0.91±0.01 | 0.33±0.02 | 0.14±0.01 |
| TCR-dWAE (null) | 1.51±0.01 | 0.15±0.01 | 0.85±0.01 | 0.46±0.06 | 0.12±0.01 |
| original | 1.59 | 0.15 | 0.92 | 0.0 | 0.14 |
| | | | RAKFKQLL* | | |
| TCR-dWAE | 1.58±0.03 | 0.08±0.0 | 0.91±0.02 | 0.43±0.07 | 0.08±0.0 |
| TCR-dVAE | 1.58±0.02 | 0.13±0.01 | 0.92±0.01 | 0.32±0.02 | 0.1±0.01 |
| TCR-dWAE (null) | 1.51±0.01 | 0.07±0.01 | 0.86±0.01 | 0.44±0.05 | 0.06±0.01 |
| original | 1.59 | 0.04 | 0.92 | 0.0 | 0.04 |
| | | | NLVPMVATV* | | |
| TCR-dWAE | 1.57±0.02 | 0.2±0.0 | 0.9±0.02 | 0.45±0.07 | 0.17±0.01 |
| TCR-dVAE | 1.59±0.02 | 0.21±0.0 | 0.92±0.02 | 0.33±0.02 | 0.17±0.0 |
| TCR-dWAE (null) | 1.51±0.01 | 0.18±0.01 | 0.86±0.01 | 0.46±0.06 | 0.15±0.01 |
| original | 1.59 | 0.08 | 0.92 | 0.0 | 0.07 |
| | | | GILGFVFTL* | | |
| TCR-dWAE | 1.56±0.02 | 0.19±0.05 | 0.89±0.02 | 0.46±0.06 | 0.17±0.04 |
| TCR-dVAE | 1.57±0.02 | 0.23±0.04 | 0.91±0.02 | 0.34±0.02 | 0.18±0.02 |
| TCR-dWAE (null) | 1.51±0.01 | 0.1±0.0 | 0.86±0.01 | 0.44±0.06 | 0.09±0.0 |
| original | 1.59 | 0.05 | 0.92 | 0.0 | 0.05 |

Table 8: Performance comparison for VDJDB, separated by peptides. Peptides with * have classification AUC < 0.8.

| | $\bar{r_v}$ | $\bar{r_b}$ | %valid | #mut/len | %positive valid |
|---|---|---|---|---|---|
| | | | MCPAS | | |
| TCR-dWAE | 1.39±0.07 | 0.29±0.03 | 0.69±0.06 | 0.47±0.03 | 0.17±0.01 |
| TCR-dVAE | 1.45±0.01 | 0.31±0.04 | 0.47±0.04 | 0.4±0.02 | 0.1±0.01 |
| greedy | 0.33±0.0 | 0.92±0.0 | 0.02±0.0 | 0.33±0.0 | 0.02±0.0 |
| genetic | 0.34±0.03 | 1.0±0.0 | 0.02±0.0 | NA | 0.02±0.0 |
| naive rm | 0.31±0.0 | 0.43±0.0 | 0.02±0.0 | 0.35±0.01 | 0.01±0.0 |
| MCTS | -0.11±0.0 | 0.94±0.0 | 0.0±0.0 | NA | 0.0±0.0 |
| TCR-dWAE (null) | 1.45±0.06 | 0.08±0.0 | 0.79±0.05 | 0.41±0.03 | 0.06±0.01 |

Table 9: Performance comparison for MCPAS, averaged across selected peptides (SSYRRPVGI, WEDLFCDESLSSPEPPSSSE, SSLENFRAYV, RFYKTLRAEQASQ, GLCTLVAML, CRVLC-CYVL)

| | VDJDB | | | MCPAS | |
|---|---|---|---|---|---|
| | valid:all | unique:valid | running time | valid:all | unique:valid |
| TCR-dWAE | 0.79±0.02 | 0.95±0.01 | 56 | 0.72±0.07 | 0.95±0.01 |
| TCR-dWAE-attn | 0.77±0.06 | 0.97±0.01 | 5553 | | |
| TCR-dWAE (null) | 0.86±0.01 | 1.0±0.0 | 56 | 0.8±0.05 | 0.99±0.0 |
| TCR-dVAE | 0.84±0.01 | 0.65±0.02 | 58 | 0.8±0.01 | 0.81±0.06 |
| greedy | 0.02±0.0 | 1.0±0.0 | 832 | 0.02±0.0 | 1.0±0.0 |
| genetic | 0.02±0.0 | 0.73±0.04 | 1389 | 0.03±0.0 | 0.71±0.08 |
| naive rm | 0.02±0.0 | 1.0±0.0 | 106 | 0.02±0.0 | 1.0±0.0 |
| MCTS | 0.0±0.0 | 0.0±0.0 | 4559 | 0.0±0.0 | 0.04±0.08 |

Table 10: Additional performance comparison. This table shows the ratio of valid sequences and unique valid sequences, as well as the running time per 5000 samples for CTPYDINQM.

| | $\bar{r_v}$ | $\bar{r_b}$ | %valid | #mut/len | %positive valid ↑ |
|---|---|---|---|---|---|
| TCR-dWAE (best) | 1.33±0.07 | 0.53±0.18 | 0.45±0.14 | 0.5±0.05 | 0.18±0.05 |
| TCR-dWAE (avg) | 1.37±0.13 | 0.51±0.21 | 0.53±0.18 | 0.46±0.09 | 0.2±0.06 |
| TCR-dWAE (random pos) | 1.36±0.03 | 0.48±0.09 | 0.61±0.06 | 0.49±0.05 | 0.23±0.02 |
| TCR-dVAE-rand (best) | 0.31±0.06 | 0.01±0.01 | 0.0±0.0 | 0.21±0.3 | 0.0±0.0 |
| TCR-dVAE-rand (random pos) | 1.44±0.02 | 0.35±0.05 | 0.45±0.04 | 0.4±0.01 | 0.1±0.01 |
| TCR-dWAE-attn (random pos) | 1.46±0.06 | 0.32±0.08 | 0.75±0.06 | 0.54±0.05 | 0.19±0.04 |
| TCR-dWAE (trimmed) | 1.35±0.08 | 0.19±0.07 | 0.68±0.08 | 0.78±0.0 | 0.11±0.02 |
| TCR-dWAE (null) | 1.51±0.01 | 0.05±0.03 | 0.85±0.01 | 0.45±0.07 | 0.04±0.03 |
| original | 1.59 | 0.01 | 0.92 | 0.0 | 0.01 |

Table 11: Comparison between model designs and modes of obtaining positive $\mathbf{z}_f$'s.

|            | ERGO        | NetTCR      |
|------------|-------------|-------------|
| TCR-dWAE   | 0.24±0.01   | 0.2±0.01    |
| TCR-dVAE   | 0.16±0.01   | 0.15±0.01   |
| greedy     | 0.02±0.0    | 0.01±0.0    |
| genetic    | 0.02±0.0    | 0.0±0.0     |
| naive rm   | 0.0±0.0     | 0.0±0.0     |
| MCTS       | 0.0±0.0     | 0.0±0.0     |

Table 12: Performance using different evaluation metrics (ERGO and NetTCR)

|                | diversity    | novelty      |
|----------------|--------------|--------------|
| TCR-dWAE       | 0.20±0.0     | 0.20±0.0     |
| TCR-dVAE       | 0.23±0.0     | 0.19±0.0     |
| VAE (collapse) | 0.14±0.01    | 0.17±0.01    |
| Validation     | 0.26         | 0.18         |

Table 13: Diversity and novelty scores for the generated sequences.

|            | 0.05          | 0.1           | 0.2          | 0.4       |
|------------|---------------|---------------|--------------|-----------|
| TCR-dWAE   | 1174.6±38.7   | 1130.4±34.0   | 460.1±28.0   | 0.1±0.1   |
| TCR-dVAE   | 809.6±49.5    | 774.6±49.8    | 290.9±25.6   | 0.0±0.0   |
| Validation | 621           | 551           | 191          | 0         |

Table 14: Number of "novel" sequences with different novelty score cutoffs.

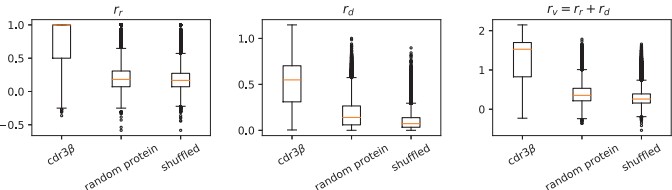

Figure 4: Distribution of TCR-AE-based evaluation metrics on known CDR3$\beta$'s, randomly selected protein segments and randomly shuffled CDR3$\beta$'s.

Figure 5: T-SNE of $\mathbf{z}_f$ and $\mathbf{z}_s$ embeddings for all peptides in VDJDB (left) and MCPAS (right). Points are colored by the label. "True" means the ground truth label. "Pred" refers to label predcited by the function classifier $\Psi$.

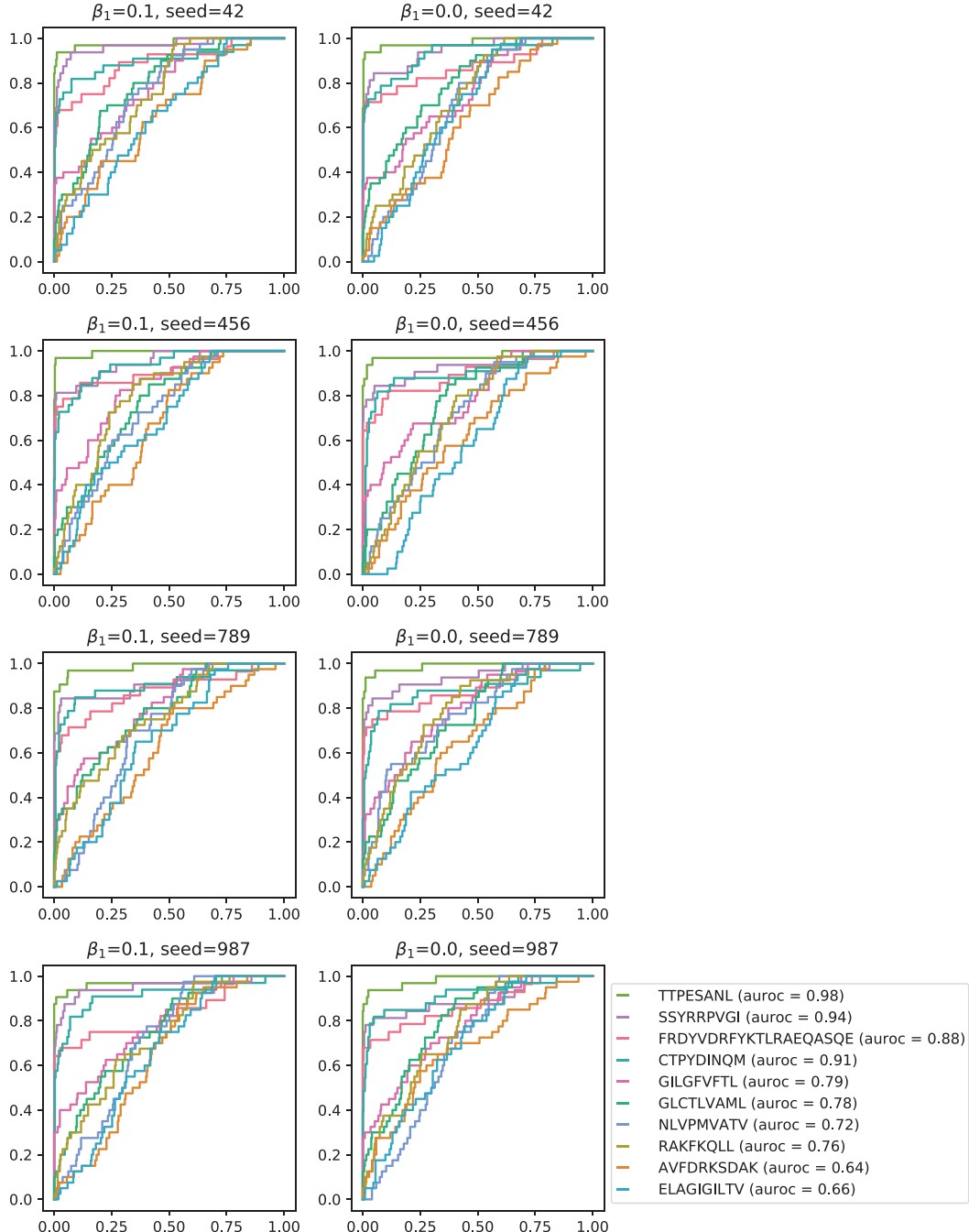

Figure 6: ROC of function classifier $\Psi$ by peptide, with different hyperparameter settings and random seeds.

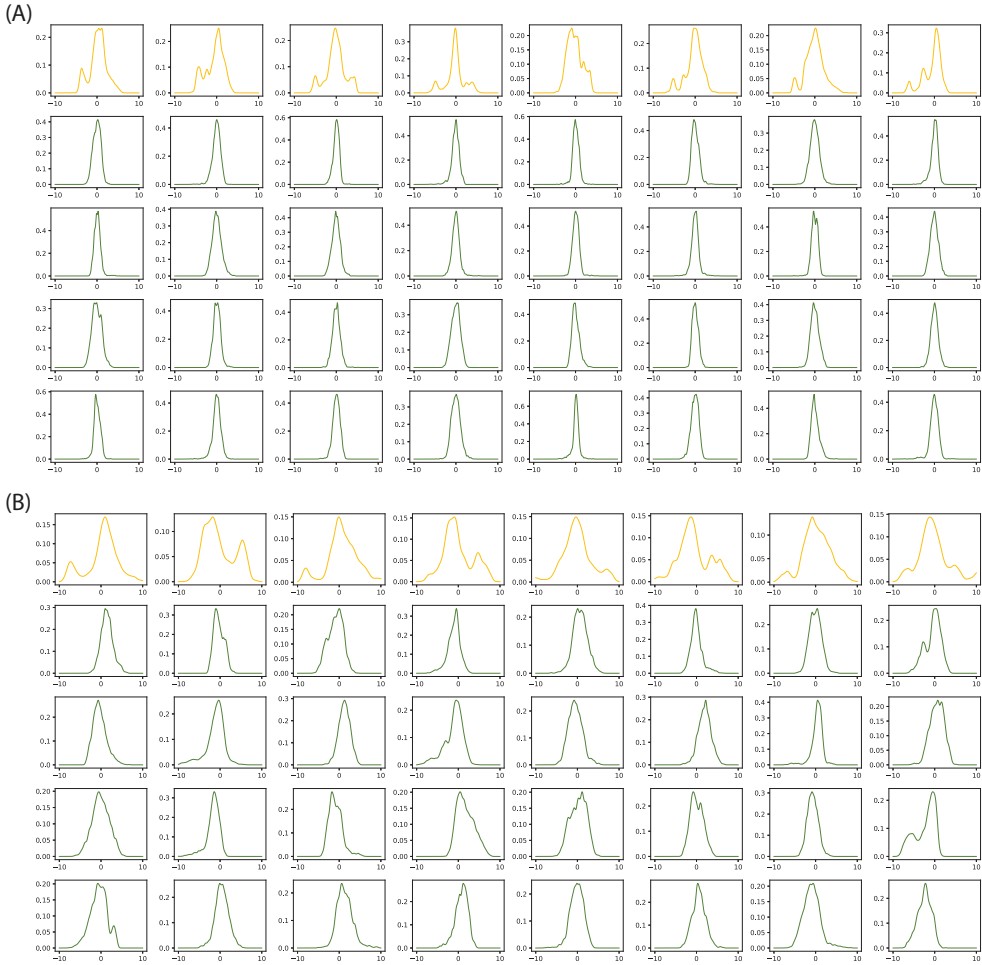

Figure 7: Distribution of the latent embeddings with (A) and without (B) Wasserstein loss. Orange lines correspond to dimensions of $\mathbf{z}_f$ and green lines $\mathbf{z}_f$. The distribution is estimated using `gaussian_kde` from the scipy package.

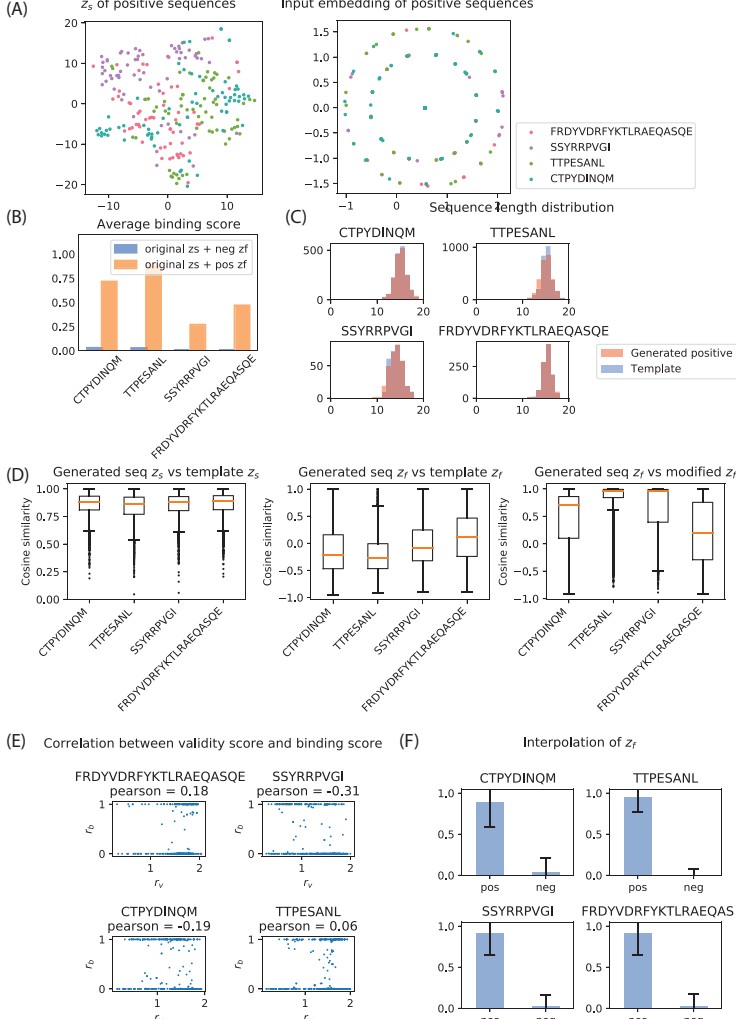

Figure 8: (A) T-SNE of $\mathbf{z}_s$ (left) and first layer embedding of the encoder (right) of positive TCRs, colored by their binding peptides. (B) The average binding score of generated positive and negative TCRs. (C) The length distribution of template and optimized TCRs (CDR3$\beta$ region) from VDJDB. (D) Cosine similarity between $\mathbf{z}_s$ of the optimized sequences vs their templates (left), $\mathbf{z}_f$ of the optimized sequences vs their templates (middle), $\mathbf{z}_f$ of the optimized sequences vs the modified $\mathbf{z}_f$ (right). (E) Scatter plot between the validity score and binding score of the engineered sequences (for simplicity, only 500 out of 5000 points are shown). (F) Binding score of interpolated samples between positive and negative pairs.

