# Appendix for "Disentangled Wasserstein Autoencoder for Protein Engineering"

## 1 Data preparation

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

| MCPAS | | | | | |
|---|---|---|---|---|---|
| | $\bar{r_v}$ | $\bar{r_b}$ | %valid | # mutations | %positive valid |
| TCR-dWAE (best) | 1.32±0.05 | 0.38±0.07 | 0.48±0.02 | 0.51±0.03 | 0.15±0.04 |
| TCR-dWAE (avg) | 1.4±0.07 | 0.31±0.03 | 0.59±0.03 | 0.44±0.03 | 0.15±0.02 |
| TCR-dWAE (random) | 1.38±0.07 | 0.29±0.03 | 0.68±0.07 | 0.47±0.03 | 0.16±0.01 |
| IDEL (best) | 1.42±0.01 | 0.34±0.07 | 0.42±0.11 | 0.43±0.03 | 0.11±0.02 |
| IDEL (avg) | 1.47±0.01 | 0.31±0.05 | 0.49±0.11 | 0.4±0.02 | 0.11±0.01 |
| IDEL (random) | 1.46±0.01 | 0.29±0.04 | 0.64±0.04 | 0.42±0.02 | 0.15±0.01 |
| greedy | 0.33±0.0 | 0.92±0.0 | 0.02±0.0 | 0.34±0.0 | 0.02±0.0 |
| genetic | 0.34±0.03 | 1.0±0.0 | 0.02±0.0 | 0.96±0.08 | 0.02±0.0 |
| naive rm | 0.31±0.0 | 0.43±0.0 | 0.02±0.0 | 0.35±0.01 | 0.01±0.0 |
| mcts | -0.11±0.0 | 0.94±0.0 | 0.0±0.0 | 0.04±0.08 | 0.0±0.0 |
| TCR-dWAE (null) | 1.45±0.06 | 0.08±0.0 | 0.79±0.05 | 0.41±0.03 | 0.06±0.01 |

Table. S4: ; Performance comparison for MCPAS, averaged across selected peptides (SSYRRPVGI, WEDLFCDESLSSPEPPSSSE, SSLENFRAYV, RFYKTLRAEQASQ, GLCTLVAML, CRVLC-CYVL)

| | VDJDB | | MCPAS | |
|---|---|---|---|---|
| | valid:all | unique:valid | valid:all | unique:valid |
| TCR-dWAE-best | 0.59±0.02 | 0.69±0.1 | 0.67±0.06 | 0.72±0.05 |
| TCR-dWAE-avg | 0.63±0.03 | 0.74±0.09 | 0.74±0.07 | 0.8±0.06 |
| TCR-dWAE-random | 0.66±0.02 | 0.9±0.02 | 0.72±0.07 | 0.95±0.01 |
| TCR-dWAE-null | 0.86±0.01 | 0.99±0.0 | 0.8±0.05 | 0.99±0.0 |
| IDEL-best | 0.73±0.02 | 0.73±0.15 | 0.76±0.0 | 0.56±0.14 |
| IDEL-avg | 0.78±0.02 | 0.76±0.14 | 0.81±0.01 | 0.61±0.14 |
| IDEL-random | 0.78±0.02 | 0.83±0.07 | 0.8±0.01 | 0.81±0.06 |
| greedy | 0.02±0.0 | 1.0±0.0 | 0.02±0.0 | 1.0±0.0 |
| genetic | 0.03±0.02 | 0.74±0.04 | 0.03±0.0 | 0.71±0.08 |
| naive rm | 0.03±0.0 | 1.0±0.0 | 0.02±0.0 | 1.0±0.0 |
| mcts | 0.0±0.0 | 0.0±0.0 | 0.0±0.0 | 0.04±0.08 |

Table. S5: Additional performance comparison. This table shows the ratio of valid sequences and unique valid sequences, as well as the running time.

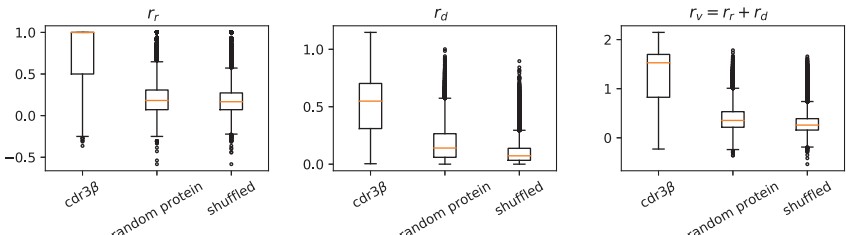

Fig. S1: Distribution of TCR-AE-based evaluation metrics on known CDR3$\beta$'s, randomly selected protein segments and randomly shuffled CDR3$\beta$'s.

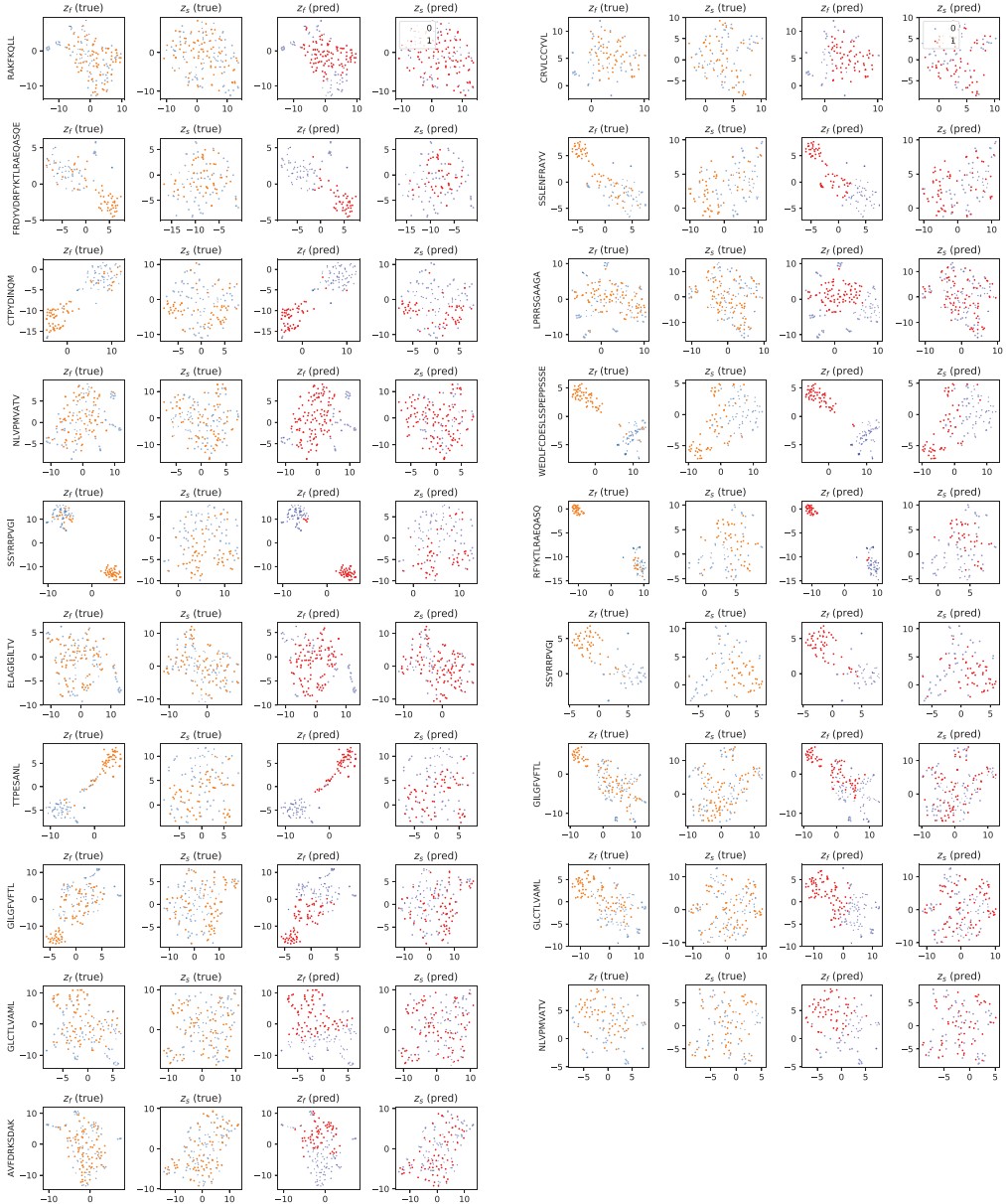

Fig. S2: T-SNE of $\mathbf{z}_f$ and $\mathbf{z}_s$ embeddings for all peptides in VDJDB (left) and MCPAS (right). Points are colored by the label. "True" means the ground truth label. "Pred" refers to label predcited by the function classifier $\Psi$.

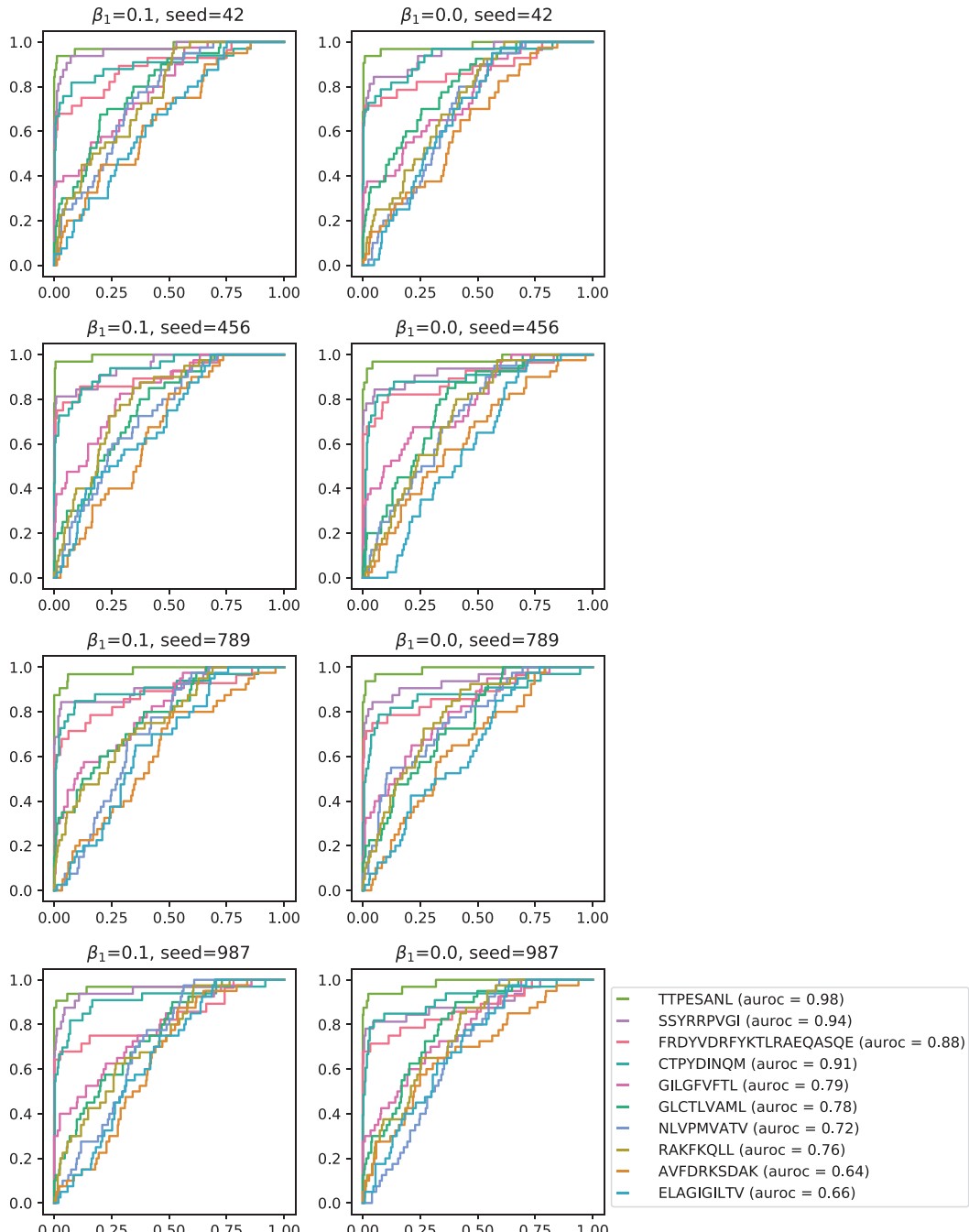

Fig. S3: ROC of function classifier $\Psi$ by peptide, with different hyperparameter settings and random seeds.

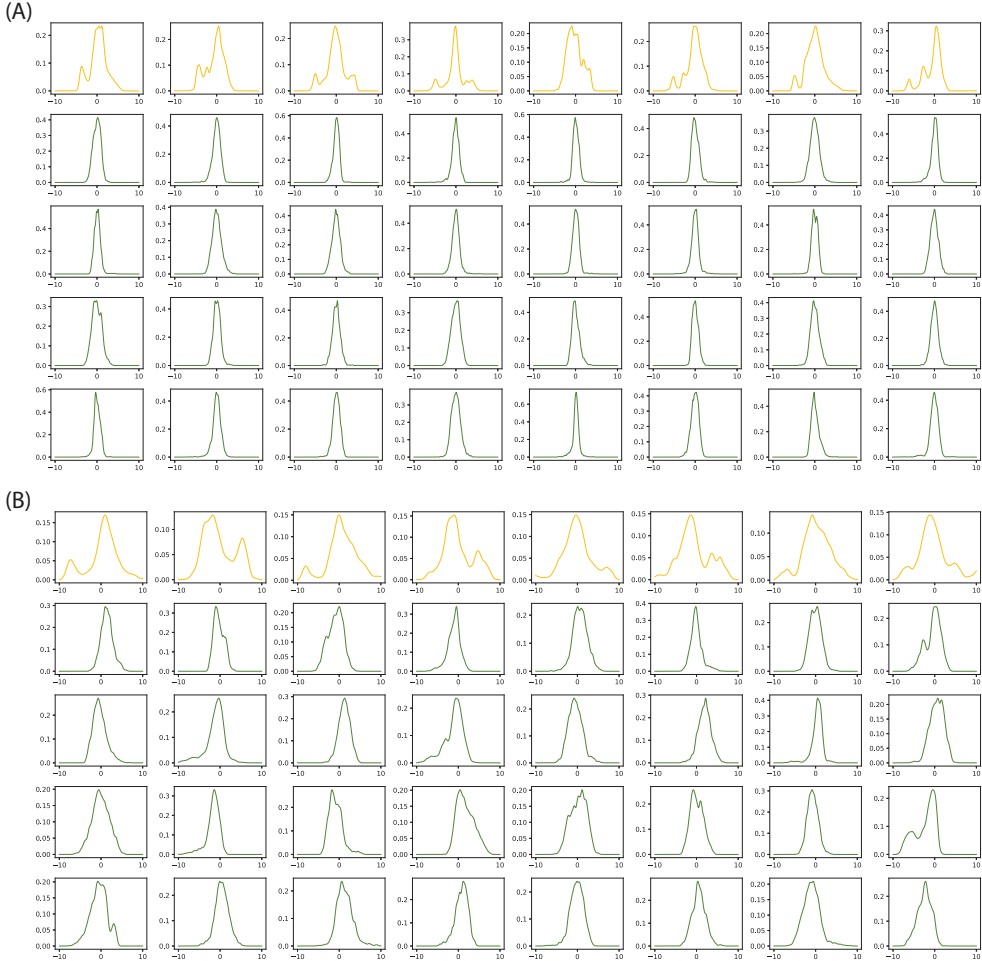

Fig. S4: Distribution of the latent embeddings with (A) and without (B) Wasserstein loss. Orange lines correspond to dimensions of $\mathbf{z}_f$ and green lines $\mathbf{z}_f$. The distribution is estimated using `gaussian_kde` from the scipy package.

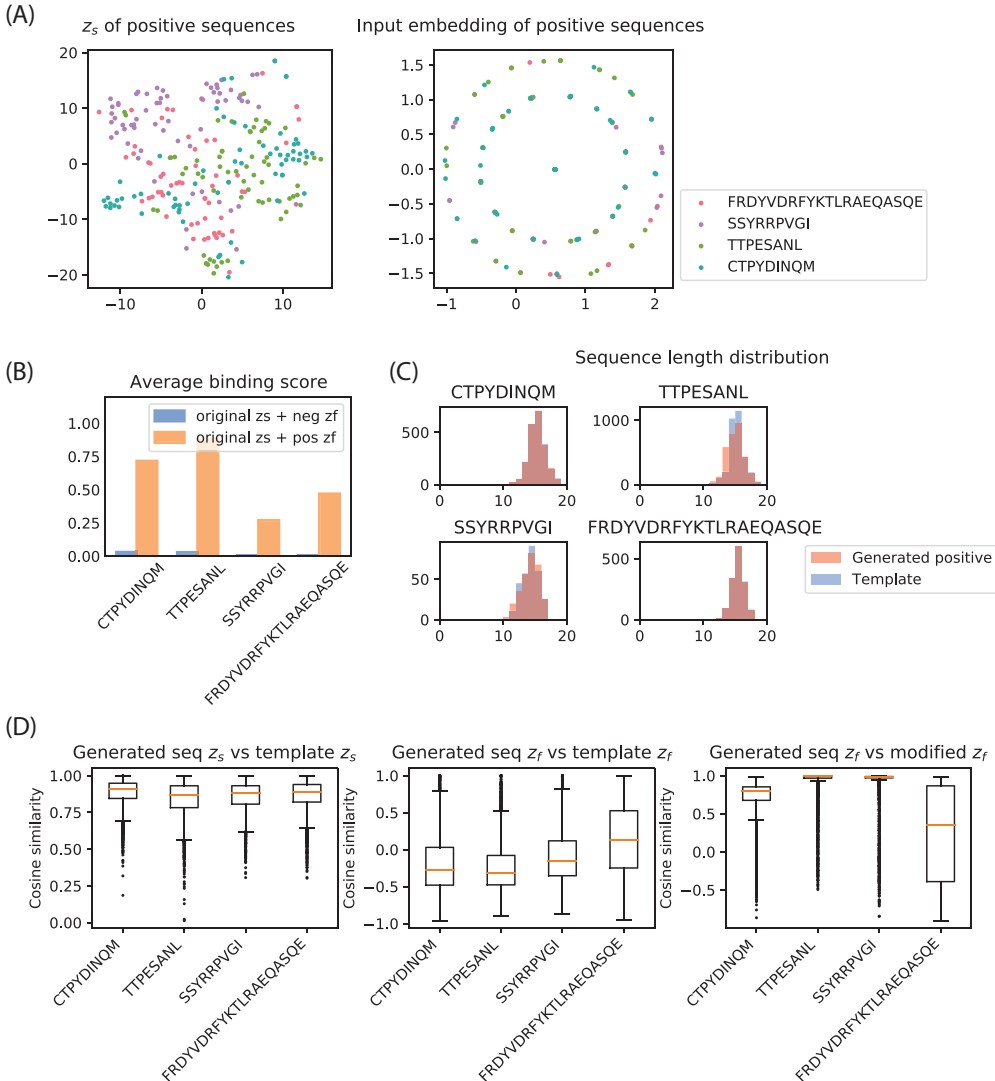

Fig. S5: (A) T-SNE of $\mathbf{z}_s$ (left) and first layer embedding of the encoder (right) of positive TCRs, colored by their binding peptides. (B) The average binding score of generated positive and negative TCRs. (C) The length distribution of template and optimized TCRs (CDR3$\beta$ region) from VDJDB. (D) Cosine similarity between $\mathbf{z}_s$ of the optimized sequences vs their templates (left), $\mathbf{z}_f$ of the optimized sequences vs their templates (middle), $\mathbf{z}_f$ of the optimized sequences vs the modified $\mathbf{z}_f$ (right).