# OpenReview forum: "Disentangled Wasserstein Autoencoder for T-Cell Receptor Engineering"
_NeurIPS.cc/2023/Conference — NeurIPS 2023 poster_

### Official Review · Reviewer_4twu · 2023-06-25

**Soundness:** 1 poor
**Presentation:** 2 fair
**Contribution:** 2 fair
**Rating:** 5
**Confidence:** 4

**Summary:**

The paper proposes to disentangle the structural from functional representations when performing machine learning (ML) assisted protein engineering. The authors propose a Wasserstein autoencoder with auxiliary classifier to learn disentangled embeddings for the structural and functional components. Once disentangled embeddings are learned, they generate new sequences with several strategies: randomly selecting positive sequences, averaging positive sequences, and selecting the sequence with the highest classifier prediction. The method is empirically validated on TCR binder generation of modifying a TCR sequence to stay as a TCR while binding to a peptide of interest. The method is compared against a few baselines which their method is able to beat on their metrics.

Their contribution is learning disentangled structure and function representations to be utilized in TCR engineering.

**Strengths:**

- To the best of knowledge, this is the first exploration of wasserstein autoencoders for the task of protein engineering. The desire of separating functional and structural knowledge and solely optimizing functional properties of a protein is a fundamental challenge.

- The disentanglement measurement is well-formulated and intuitively makes sense. The training objective is justified in this regard of maximizing disentanglement measurement.

**Weaknesses:**

- Implicit vs. explicit disentanglement. The authors should be careful in stating their method is the "first approach that utilizes disentangled representations for protein engineering" (line 15). Their method is a form of implicit disentanglement where the learned embeddings are separated to have a structural and functional component. Prior works such as SMCDiff [1], RFdiffusion [2], constrained hallucination [3] and Cao et al [4] perform explicit disentanglement where one part of the protein is kept fixed and the remaining sequence is optimized. Here the structure is assumed to be known a priori therefore the methods can directly keep fixed the part of the protein not necessary for function. A discussion around the comparison between these two views of the problem is necessary. Additionally they should have benchmarked against the motif-scaffolding methosd if the authors pick TCR for which the structure is very well known. i.e. [5, 6] could ( and should) have been benchmarks.

- Only benchmarking on TCR. I do not find the experiments compelling to support the claim of separating the functional and structural components. TCR is well-studied and lots of data exists. We know very well there is a highly conserved region and only the CDR regions vary depending on the binding target. Therefore TCR seems like a easy task to separate the functional and structural components from the evolutionary data. Perhaps a toy example where the method can detect non-trivial disentanglement would be helpful or another protein system? I can also think of another baseline of only limiting the mutations to the CDR regions. How would this fare against the proposed method?

- Claim of preserving structural backbone. The authors seem to conflate sequence vs. structural backbone conservation. They utlize a sequence classifier to classifier if a sequence is TCR or not. That is fine but this is not the same as keeping the structural backbone the same. What does it mean to stay as a TCR? This definition is not provided. How do we know the structural backbone is conserved based on the classifier predictions? The sequences are not ran through AlphaFold2 which could have supported their claim but we also know AF2 is not sensitive to point mutations. The notion of "structural embedding" (line 29) used throughout is misleading because it has no relationship to structure based on what is presented.

- Lack of related work. I find it concering there is little mention of NetTCR throughout the work. As I understand it, NetTCR is a critical comparion  the function classifiers and oracles used in this work. In addition there is no discussion of machine learning methods applied to antibody engineering. The lack of this discussion makes the method appear more novel than it is.

- Missing method and metric clarity. There are missing definitions of symbols (see questions) and some decisions are made arbitrarily. For instance, binding score is positive if $\psi(z_f`, u) > 0.5$ where 0.5 is the operating point but why 0.5? There needs to be justification for this operating point. Otherwise the operating point could've been cherry picked to make the method look better.





[1] https://arxiv.org/abs/2206.04119
[2] https://www.biorxiv.org/content/10.1101/2022.12.09.519842v1
[3] https://www.science.org/doi/10.1126/science.abn2100
[4] https://www.nature.com/articles/s41586-022-04654-9
[5] https://arxiv.org/abs/2207.06616
[6] https://www.biorxiv.org/content/10.1101/2022.07.10.499510v1
[7] https://www.nature.com/articles/s42003-021-02610-3

**Questions:**

- As stated in weaknesses, why is there no direct comparison to NetTCR and other antibody engineering methods? I imagine the antibody engineering methods could readily be applied here?

- In line 121 what is beta_1 and beta_2? What are they set to?

- What is $\Sigma$ on line 117?

- Line 110-111. Why does making the embedding space matc N(0,1) lead to independence of z_f and z_s? For someone not familiar with Wasserstein autoencoders, this needs to be explained better. Additionally what is the meaning of "gaussian shape of the latent space facilitates generation" on the same line. I thought the method is deterministic so why is this mentioned? Is there sampling happening?

- Line 160-151. What is meant by "well-classified by ERGO"?

- Again I am confused why the authors went out to develop their own classifiers if ERGO and NetTCR are state-of-the-art. Did these methods have a limitation that is not mentioned?

- Line 164-165. "We generate an additional dataset with 5000 TCR sequences that do not overlap with the training set." How are the sequences generated and what is the definition of "does not overlap"?

- Line 176-177. For generating positive z_f`, one of the ways is to use the best z_f that produces the highest classifier prediction. How is this generation done? Why wouldn't this lead to adversarial samples? For "random" I am unclear of the motivation of this technique. Surprisingly it does very well compared to the other methods. Why does that happen?

- Line 185-186. Why is a autoencoder used as a oracle? It seems very odd to use a continuous latent space for a discrete generation task. Why is a transformer or LSTM not used?

- Comparison to IDEL. Since IDEL is a VAE, shouldn't the authors generate multiple samples or do some sort of greedy search in its latent space for a fair comparison?

- Line 207. It sates the majority of generated valid sequences are unique and novel. What is meant by unique and novel? Is there a sequence identity cutoff? I see no discussion of this in the appendix though the appendix is referenced.

**Limitations:**

The authors only explain one limitation which is their experiments are performed on a smaller subset selected by ERGO. I don't believe there is sufficient analysis of the failure modes of their method. Why and when does the method fail? Are there examples theirs does good on where baselines fail? This would shed light on the limitations but it is missing.

---

> ### Author Rebuttal · Authors · 2023-08-10
>
> Thanks for your comments. Here we respond to your points and questions in the sequential order:
>
>  **1. Explicit and implicit**
>
> By explicit and implicit, here we are talking about the nature of the latent space. For some generative models, such as diffusion models,  there is no “explicit” latent space. Methods like VAE and WAE “explicitly” model the distribution of the latent space, so we could directly enforce disentanglement on that latent space.
>
> On the other hand, the “explicit” you mean with the examples is manually defining functional and structural parts of the sequence. There is no such prior knowledge for the CDR3beta region. Thus, we need to learn self-supervised disentanglement on an **explicit latent space** because we cannot do such separation manually and “explicitly”.
>
> Specifically, motif scaffolding is a reverse problem of ours. It extends a functional “motif” by generating a “scaffold”, while we seek to modify the function of an existing TCR sequence. It also requires prior knowledge of the functional part, while in our case it needs to be learned from the data. Thus, it is not directly comparable here.
>
>
> **2. Usage of TCR**
>
> In fact, we actually are focusing on the CDR3beta region, as mentioned in 2.1. We also find that the function/structure separation is non-trivial within CDR3beta as the model learns patterns all across the region.
>
> **3. Structural backbone**
>
> We acknowledge that our study is limited by the lack of high-quality TCR structures or reliable prediction methods. Thus, we can only estimate structural similarity indirectly through the embedding patterns of the sequence, based on the intuition that the sequence defines the structure.
>
> **4. Related work; NetTCR**
>
> NetTCR is a classifier. It cannot be directly used for sequence engineering.
>
> Most computational antibody engineering methods either use physical modeling, which is slow on a large scale, or rely on 3D structural data, which is scarce for TCR. Thus, they are not comparable to ours.
>
> **5. 0.5 as the cutoff**
>
> To select positive $z_f$’s in "random" and "avg" modes, we actually do not use the classifier $\Psi$ (which is used only in "best" mode). Instead, we directly use the $z_f$’s  of the sequences with positive labels from the dataset. The information in 3.2.1 is inaccurate. Thanks for pointing out. We will correct it in the next version.
>
> **6. Symbols**
>
> Beta_1 and beta_2 are the weights of the losses. The values are 1.0 and 0.1, as mentioned in Appendix 2.2. Σ means sum. We will further clarify it in the manuscript.
>
> **7. WAE and the latent space**
>
> WAE models the distribution of the latent space by enforcing a shape on it. The training and decoding are deterministic, but it still has a latent space distribution that could be sampled from.
>
> For our experiments, however, we manipulate existing embeddings and deterministically decode them to generate new samples, **with no random sampling involved**, which is different from VAEs.
>
> The Gaussian distribution facilitates such generation because it has a regular shape. If one manipulates  (Section 3.1) a latent embedding, the resulting “new” embedding is highly likely to remain in the distribution and thus could be decoded properly. With an irregular shape, it may fall out of the distribution leading to poor-quality generations.
>
>
> **8. Well-classified by ERGO**
>
> We have explained the selection process in Appendix 1.2. We will also clarify it in the main text.
>
> **9.“Develop our own classifiers”**
>
> ERGO and NetTCR all take the **raw sequence** as the input. In order to manipulate the latent space, however, we’ll need a classifier on the **latent embeddings**. It is used as an auxiliary to control the content of the embeddings.
>
> **10. Additional dataset**
>
> We select 5000 additional sequences from VDJDB which are not seen in ("do not overlap with") the training set.
>
> **11."Best" and “random” mode**
>
> For the “best” method we obtain the $z_f$’s from the sequence in the dataset with the highest binding score from $\Psi$. Also, we’re not sure why the reviewer believes this would lead to adversarial samples.
>
> As stated in 3.2.1, the “random” method uses the $z_f$ from a random **positive** sample. With a well-regularized latent space, any positive $z_f$ should carry patterns related to positive binding. Thus the high performance of “random” is expected. For better clarity, we will rename it to "random pos".
>
> **12. Using the autoencoder, not LSTM or transformer**
>
> The goal here is to quantify how likely the generated sequence comes from the same distribution of known TCRs (i.e. valid), similar to autoencoder-based out-of-distribution detection (OOD) methods (refs below). For such a task, continuous-valued embeddings from an autoencoder are more computationally efficient. We show in Appendix Fig. S1 that these metrics can separate real TCRs from random protein segments.
>
> Also, transformers and LSTMs can be used in many model architectures, including autoencoders. In fact, we are using **LSTM-based autoencoder** here as is mentioned in Appendix 4.1.
>
> **13. Comparison to IDEL**
>
> Thanks for bringing this up.  We have re-done the experiment with 5 random samples for each embedding, and choose the one with the highest binding score.
>
> **14. Meaning of unique and novel**
>
> Sorry for the confusion. We’re looking for literally “unique” and “novel” sequences through direct sequence comparison, and there is no sequence similarity involved.
>
> **15. Failure cases**
>
> Our method would fail when:
>
> (1) There is no sufficient data.
>
> (2) The functional label is not well-defined or well-classified (e.g. the poorly classified peptides we do not include in Table 1),  or when there is no inherent (yet unknown) separation between the features.

---

> > ### Comment · Reviewer_4twu · 2023-08-14
> > **Response**
> >
> > Thank you for answering my questions. I have read the global rebuttal and the author's rebuttal.
> >
> > >  explicit latent space because we cannot do such separation manually and “explicitly”... Specifically, motif scaffolding is a reverse problem of ours.
> >
> > I disagree. Recognize that motif-scaffolding is a general framework for binder design. One can set the fixed region to be the non-CDR region and the design region to be the CDR. **The work claims to be discovering the functional/structural latent but the latent is "manually" limited to be the CDR, which is known to confer the functional properties.**  The manuscript acknowledges this in two locations:
> > * Lines 19-20, "the separation of the overall structure and the smaller 'functional' site, such as the .. immunoglobulin fold versus the antigen-binding complementarity-determining region (CDR) in immunoproteins"
> > * Lines 78-79, "Following [23–25], we here limit our modeling to the CDR3β region since it is the most active region for TCR binding."
> >
> > The claims would be stronger if the method discovered the functional regions without a priori limiting the modeling space.
> >
> > > 4. Related work; NetTCR
> >
> > Apologies for not being clear. My question is why the metrics were obtained using ERGO rather than NetTCR? The latter was published in 2021 and showed clear improvement over ERGO.
> >
> > > 7. WAE and the latent space... For such a task, continuous-valued embeddings from an autoencoder are more computationally efficient. We show in Appendix Fig. S1 that these metrics can separate real TCRs from random protein segments.... we’re not sure why the reviewer believes this would lead to adversarial samples.
> >
> > I'm combining several of the author's comments here. A continuous latent space gives up the discrete structure of the problem. It has been documented in that optimizing in a (continuous) latent space leads to adversarial examples since much of the latent space is empty. See section 3 of [1], "Although in principle optimization can be performed over all of Z, it has been widely observed that optimizing outside of the feasible region tends to give poor results, yielding samples that are low-quality, or even invalid (e.g. invalid molecular strings, non-grammatical sentences)"
> >
> > As the authors showed in figure S1, the method is able to distinguish between CDRs and random sequences. However, I would expect the disadvantages of latent space optimization to be a larger issue in higher dimensional spaces where structure needs to be imposed, i.e. [2].
> >
> > > 14. Meaning of unique and novel
> >
> > Please see [3] where the diversity and novelty metrics are defined. A median score of the levenshtein distance is appropriate since there is redundancy in the residue identities. A different of just one residue will usually not amount to a meaningful difference.
> >
> >
> > [1] Tripp, Austin, Erik Daxberger, and José Miguel Hernández-Lobato. "Sample-efficient optimization in the latent space of deep generative models via weighted retraining." Advances in Neural Information Processing Systems 33 (2020): 11259-11272.
> >
> > [2] Maus, Natalie, et al. "Local latent space bayesian optimization over structured inputs." Advances in Neural Information Processing Systems 35 (2022): 34505-34518.
> >
> > [3] Jain, Moksh, et al. "Biological sequence design with gflownets." International Conference on Machine Learning. PMLR, 2022.

---

> > > ### Author Response · Authors · 2023-08-17
> > >
> > > Thanks for the reply. We are happy to revise our paper to clarify our limitations and accurately demonstrate our contributions following your suggestions.
> > >
> > > **1. Explicit and implicit**
> > >
> > > We believe full TCR design especially motif scaffolding is impractical for our setting because:
> > >
> > > 1. Structural data is scarce. The largest database for TCR-pMHC 3D structures to our knowledge [1] contains only 694 entries from 123 non-redundant PDB records.
> > >
> > > 2. Motif scaffolding requires a fixed “functional” motif, and designs the scaffold to support it. In our case, we seek to **design the motif itself** instead. Motif scaffolding is its downstream task, and we don’t have sufficient data for that.
> > >
> > > Indeed, we are learning a “structural/functional” pattern separation **within the CDR3beta**. The word “structural” here means the sites that must be preserved to maintain the validity. These should be separated from the sites that are “safe” to edit and could impact the function, i.e. the “functional” patterns.
> > >
> > > Also, based on you and other reviewers’ comments, we have limited our claim, including the title, to “TCR design” not protein.
> > >
> > > [1] Borrman, Tyler, et al. "ATLAS: a database linking binding affinities with structures for wild‐type and mutant TCR‐pMHC complexes." Proteins: Structure, Function, and Bioinformatics 85.5 (2017): 908-916.
> > >
> > > **2. NetTCR**
> > >
> > > Thanks for the clarification.
> > >
> > > NetTCR only offers predictions for three peptides, while we perform the study on the entire VDJDB. Thus, ERGO offering a prediction score to more peptides suits our task better.
> > >
> > > For the comparison, we re-train a one-chain NetTCR on our own training set with more peptides. The evaluation result on the generated sequences in Table 1 in the manuscript (reporting the *%positive valid* only) with the two classifiers is as follows. The conclusion does not change. We will add these results to the appendix.
> > >
> > >
> > > |                   | ERGO      | NetTCR    |
> > > |:------------------|:----------|:----------|
> > > | TCR-dWAE (random) | 0.24±0.01 | 0.2±0.01  |
> > > | IDEL (random) | 0.16±0.01 | 0.15±0.01 |
> > > | greedy            | 0.02±0.0  | 0.01±0.0  |
> > > | genetic           | 0.02±0.0  | 0.0±0.0   |
> > > | naive rm          | 0.0±0.0   | 0.0±0.0   |
> > > | MCTS              | 0.0±0.0   | 0.0±0.0   |
> > >
> > > It is also worth mentioning that Fig. 2 in the NetTCR 2.0 paper shows it has no clear advantage over ERGO **when using CDR3beta only**.
> > >
> > > **3. Unique and novel**
> > >
> > > Thanks for your suggestions. We calculate the novelty score following [1] for *each individual sequence* from the *positive valid* subset in Table 1:
> > >
> > > $\text{Nov}(x) = \min_{s_i \in D_0} d(x, s_i)$
> > > where $D_0$ is the set of templates and the distance measure $d$ is the levenshtein distance relative to the length of the longer sequence.
> > >
> > > This score can then be used to select the “novel” sequences. We also include a held-out validation set as a reference. We find that around 1/3~1/2 of the sequences remain at cutoff 0.2 and all are excluded at cutoff 0.4. This follows a similar pattern as the validation set.
> > >
> > > |                  | 0.05        | 0.1         | 0.2        | 0.4     |
> > > |:----------------|:------------|:------------|:-----------|:--------|
> > > | IDEL            | 809.6±49.5  | 774.6±49.8  | 290.9±25.6 | 0.0±0.0 |
> > > | dWAE            | 1174.6±38.7 | 1130.4±34.0 | 460.1±28.0 | 0.1±0.1 |
> > > | Validation*     |  621.0  | 551.0 | 191.0 | 0.0 |
> > >
> > > Here we also report the same novelty and diversity metrics for the positive valid subset as in [1]. We would like to point out that:
> > >
> > > (1) These metrics evaluate the set of generated sequences as a whole, while we directly select the sequences that meet the validity and binding score criterion, so they are not our primary objective.
> > >
> > > (2) The comparisons are only meaningful within different generative models. For the random mutation-based methods, one can always achieve high novelty and diversity by introducing a sufficient number of mutations.
> > >
> > > We show that the results are not very different between TCR-dWAE and IDEL. In addition, we include the case of a VAE which suffers from posterior collapse (where the generation is always limited to the same few patterns). The scores for the first two are at a similar level as the validation set, while the diversity score for the collapsed VAE drops a lot. These suggest that the positive valid sequences presented in the paper are sufficiently diverse and novel.  We will incorporate these results into the paper.
> > >
> > > |                   | diversity   | novelty   |
> > > |:------------------|:------------|:----------|
> > > | TCR-dWAE (random) | 0.2±0.0     | 0.2±0.0   |
> > > | IDEL (random)     | 0.23±0.0    | 0.19±0.0  |
> > > | VAE (collapse)   | 0.14±0.01   | 0.17±0.01 |
> > > | Validation*   | 0.26   | 0.18  |
> > >
> > > [1] Jain, Moksh, et al. "Biological sequence design with gflownets." International Conference on Machine Learning. PMLR, 2022.

---

> > > > ### Author Response · Authors · 2023-08-17
> > > >
> > > > (Continued)
> > > >
> > > > **4. Continuous latent space**
> > > >
> > > > We agree with your points on latent space optimization (LSO), but please realize that this is not at all the focus of our paper. Our main goal is to regularize and operate on the latent space to facilitate manipulations for a specific biological application. We are **not** proposing “continuous LSO for discrete data” as our innovation because it has been an established common practice for discrete sequences such as texts and proteins, as described in [1~6], including [1] you cited, which is about a training method for “optimization in the **low-dimensional, continuous latent manifold**” (abstract) and uses VAE, a latent space model, for all three tasks (section 6).
> > > >
> > > > Furthermore, we do not feel that our application suffers particularly from the limitations of LSO that you point out. In fact, the comparable performance of multiple ways of LSO (best/avg/random) in Table 1 suggests a rather good regularization of our latent space, and a generally successful avoidance of infeasible regions, otherwise some of these would be significantly worse. We do agree our approach can be further enhanced with various new techniques that improve the latent space, and we leave that for future work.
> > > >
> > > >
> > > > [1] Tripp, Austin, Erik Daxberger, and José Miguel Hernández-Lobato. "Sample-efficient optimization in the latent space of deep generative models via weighted retraining." Advances in Neural Information Processing Systems 33 (2020): 11259-11272.
> > > >
> > > > [2] Lample, Guillaume, et al. "Multiple-attribute text rewriting." International Conference on Learning Representations. 2018.
> > > >
> > > > [3] Fu, Zhenxin, et al. "Style transfer in text: Exploration and evaluation." Proceedings of the AAAI Conference on Artificial Intelligence. Vol. 32. No. 1. 2018.
> > > >
> > > > [4] John, Vineet, et al. "Disentangled representation learning for non-parallel text style transfer." arXiv preprint arXiv:1808.04339 (2018).
> > > >
> > > > [5] Cheng, Pengyu, et al. "Improving disentangled text representation learning with information-theoretic guidance." arXiv preprint arXiv:2006.00693 (2020).
> > > >
> > > > [6] Castro, Egbert, et al. "Transformer-based protein generation with regularized latent space optimization." Nature Machine Intelligence 4.10 (2022): 840-851.

---

> > > > > ### Comment · Reviewer_4twu · 2023-08-21
> > > > > **Response**
> > > > >
> > > > > Thank you for the reply.
> > > > >
> > > > > > 1. Explicit and implicit
> > > > >
> > > > > After reading I think our disagreement comes from semantics of "structural/functional" patterns as presented in your introduction. I would highly recommend changing this wording in the introduction and instead formulate it as simply **non-functional** and **functional** which I think you are getting at. Structure confers function so it can lead to a knee-jerking negative reaction from structural biologists.
> > > > >
> > > > > Still I believe the techniques of motif-scaffolding are applicable here even with little data available when one only designs the CDR region. I do not mean it needs to be compared against here but it is still a viable method that has not been disproven.
> > > > >
> > > > > > 3. Unique and novel
> > > > >
> > > > > Thank you for including there. I agree it is not your primary objective and I do not mind if your method does not beat other methods. I believe it makes the results more comprehensive to include these and characterize areas for possible improvement.
> > > > >
> > > > > > 4. Continuous latent space
> > > > >
> > > > > Thank you for the discussion. I like the characterization of your method as regularizing the latent space.
> > > > >
> > > > > I have raised my score from 3 to 5. I'm unable to go higher because my score has to partially reflect the initial issues with the manuscript which we cannot see the updated version of. Overall, I recommend acceptance.

---

### Official Review · Reviewer_TMVT · 2023-06-29

**Soundness:** 4 excellent
**Presentation:** 4 excellent
**Contribution:** 3 good
**Rating:** 7
**Confidence:** 5

**Summary:**

In many protein engineering problems, we want to maintain the overall fold of the protein while changing its functional properties. This is analogous to style transfer in computer vision. The paper uses a disentangled Wasserstein autoencoder to learn separate functional and structural embeddings for T-cell receptors (TCR). First, the authors show theoretically that their model should isolate peptide-binding information from information about the TCR scaffold structure. Next, they show that feeding their model new target peptide embeddings results in new TCR sequences that are predicted to be valid and to bind to the target peptide. Finally, they show empirically that the representations are indeed disentangled and demonstrate via ablation studies that all aspects of their model are required for strong performance.

**Strengths:**

The paper describes an original, performant solution to a significant problem in protein engineering. I found it difficult to believe that similar methods have not been previously applied to protein engineering, but after spending some time searching, that does appear to be the case. In general, the exposition is clear. The experiments are thorough and provide strong evidence that their solution is better than baseline methods and that all components of their model are required for performance.

**Weaknesses:**

## Major

The title and introduction make general claims about "protein engineering," but the remainder of the paper focuses on TCRs, and specifically on a short segment called the CDR3Beta region. The authors should either limit the claims to match their experiments, or add experiments to show that this framework will work for protein engineering in general.

In TCRs, binding is mostly mediated by the variable region within the CDR, and indeed the model learns to mostly change that region, as shown in Figure 2a and b. The results would be a lot more general if the model were applied to a case where there is not such a clean split between the functional region and the scaffold. This also leads to the question of whether the model needs to learn how to disentangle function and scaffold from data. Two possible experiments to validate this:
1. Only model the hypervariable region and use a constant consensus sequence for the first 4 and last 3 residues
2. Separately embed the hypervariable and conserved regions directly, instead of having the model learn them from the data.

In addition, there are other works that explicitly hold the functional part of a protein sequence or structure constant and generate a scaffold around it. See, for example, https://www.biorxiv.org/content/10.1101/2022.12.09.519842v1 and https://www.nature.com/articles/s43588-023-00440-3. This is the reverse of what this paper tries to do, so I don't think it's necessary to do direct comparisons, but it should be mentioned in the related work.

## Minor

Table 1 is difficult to interpret. As their primary empirical result is that no baseline matches their method on both validity and binding, it could be clearer to plot generated sequences by validity and binding colored by method to show that the sequences from their method lie at the pareto front.

It's unclear to me how much the theorem actually strengthens the paper, or how it's used in the architecture or results. As the authors state right after the theorem, the independent Gaussian prior and classification loss already promote disentanglement.

It's a little confusing that the "structural embedding" is never used to predict a structure, only the CDR sequence.

In Figure 3c, it would be nice to label the embedding components directly on the figure in addition to describing them in the legend.


**Questions:**

How well would this method work on more general, larger, proteins where there's a less obvious decomposition between scaffold and function in sequence space?

Alternately, for TCRs, how much better is this method than explicitly decomposing the scaffold and function in sequence space?

Why use an LSTM decoder? There's already a transformer encoder -- wouldn't it be easier to also use a transformer decoder?

In general, very little information is given about model and training hyperparameters, including model size, hardware used, and length of training.

**Limitations:**

As is, while promising, the paper is limited by the fact that empirical results are only presented on a short segment of TCR hypervariable regions. This is not a typical protein engineering case: it only addresses one type of function (binding to peptides), in a non-typical protein, and glosses over the fact that they do not even engineer the entire protein. The method is still interesting, and I am hoping to increase my score (a little) if the authors clearly limit their claims and state limitations, or (a lot) if they demonstrate that this works more generally.

---

> ### Author Rebuttal · Authors · 2023-08-10
>
> Thanks for your comments. Here we respond to your points and questions in the sequential order:
>
> **1. Limited scope**
>
> We focus on TCR here because it is easier to define a quantifiable “function” (binding vs non-binding). For many other scenarios, such a definition might be ambiguous. Also, there is quite a large amount of labelled TCR-peptide pairs that are sufficient for machine learning, while for other applications it may require careful data collection and curation. We would like to leave them for future studies.
>
> We do agree that the claim on “protein engineering” as a whole is too broad for the paper. Thus, we will limit our claim to the engineering of TCR CDR3beta and modify the title, introduction and conclusion accordingly.
>
> **2. Conservative and hypervariable regions**
>
> We observe that the structural and functional patterns learned by the model could span the entire sequence, as modifying $z_f$ could potentially lead to modifications in both the conservative region and the hypervariable region (Fig 2B and Attached Fig B), and that the functional and structural encoders attend to some overlapping positions, which does not correspond to the boundary of hypervariable/conservative regions (Attached Fig A). Thus, a manual cut between a “functional” hypervariable region and a “structural” conservative region may not be a good idea.
>
> **3. Motif scaffolding**
>
> We very much appreciate the suggestion. We will add a paragraph to the introduction and related work on motif scaffolding.
>
> **4. Table 1; Pareto efficiency**
>
> Thanks for the suggestion. We find that the pareto efficiency analysis is less informative in our case because the predicted binding scores are rather extreme (either 0 or 1; Attached Fig. C), so the pareto front always overlaps with $r_b=1$ (Attached Fig. D) . Thus, we would still like to keep Table 1 as we feel that the count of valid binders from a hard cutoff on both scores, rather than the scores themselves, is a better evaluation criterion. We will also improve its readability later.
>
> **5. Theorem 2.1**
>
> From a theoretical perspective, the disentanglement we want to achieve has two parts:
>
> 1. Independence, or minimal mutual information, between the dimensions. This is indeed enforced by the Gaussian prior.
>
> 2. Besides independence, we also would want different parts of the embedding (i.e. $z_s$ and $z_f$) to correctly encode the desired information. Otherwise, the disentanglement would be uninformative. $z_s$ should have information about the majority of the sequence. $z_f$ should have information on some key parts of the sequence and the functional label. Thus, both embeddings need maximal mutual information with the raw sequence $x$. The theorem proves that this objective can be achieved.
>
> **6. The use of “structural embedding”**
>
> The “structure” here means “sequential patterns that define the structural backbone”. For simplicity, we use the word “structural”. However, we are also open to suggestions for more accurate wording.
>
> It is hard to predict the “structure” in the sense of protein 3D structure because the CDR3 region is a very flexible loop, and high-quality structure and reliable prediction methods are scarce. Thus, we can only rely on the available sequential information of known CDR3beta's to determine whether generated ones are valid, or whether they preserve the structural backbone, based on the intuition that the structure is defined by the sequence. If certain pivotal residues and motifs (which we try to make the “structural embedding” learn) are similar between two sequences (e.g. generated and known), they should have similar structures. We will state its limitations in the discussion section of the paper.
>
> **7. Fig. 3C**
>
> Thanks for the suggestion. We will modify the figure accordingly.
>
> **8. Generalization to proteins**
>
> That should depend on the availability and nature of data. If there is a weaker decomposition between scaffold and function (or any feature in general), or if there is not a clear definition of the function, the model may not be able to learn a meaningful disentanglement between such features (e.g. the poorly classified peptides not included in Table 1). In fact, we devise our model based on the assumption that there is some natural yet unknown decomposition within the data. We will further discuss these limitations in the discussion section of the paper.
>
>
> **9. Explicit decomposition**
>
> If you mean hand-picking the structural and functional parts of the sequence, it would be impractical on a larger scale as we can only rely on some rough heuristic rules that heavily depend on the context.  As we've shown in question 2, in our problem, the functional and structural patterns cannot be manually separated and need to be learned from the data.
>
> **10. LSTM and transformer decoder**
>
> As our dataset is not very complex, we believe an LSTM is sufficient and is easier to implement. Also, in our application, there is only one embedding for the whole sequence, while a typical transformer decoder would accept token-wise embeddings. The latter may not be easily manipulated for sequence engineering.
>
> Nevertheless, we have experimented with a slightly modified transformer decoder that accepts one single embedding. The performance as shown below is similar to LSTM, while its autoregressive sampling is much slower.
>
> |                   | $\bar{r_v} $  | $\bar{r_b}$   | %valid   | #mut/len   | %positive valid   |
> |:------------------|:------------|:----------|:-----------------|:--------------|:--------------------------|
> | TCR-dWAE (random pos) | 1.36±0.03   | 0.48±0.09 | 0.61±0.06        | 0.49±0.05     | 0.23±0.02                 |
> | TCR-dWAE-transformer  (random pos)  | 1.46±0.06   | 0.32±0.08 | 0.75±0.06        | 0.54±0.05     | 0.19±0.04                 |
>
> **11. Hyperparameters & hardware**
>
> Hyperparameters and hardware settings are specified in Appendix 2.2.

---

> > ### Comment · Reviewer_TMVT · 2023-08-10
> > **Reviewer response to rebuttal**
> >
> > Thank you for a thorough rebuttal.
> >
> > 1. I am satisfied with this, but would like to see the proposed revisions.
> > 2. In the general rebuttal, the authors state
> > >These observations indicate that both the structural and functional patterns the model learns are higher-level patterns and could span both the conservative and hypervariable regions, and cannot be fit into manually defined windows. Thus, we believe a manual separation or exclusion of certain residues is not possible, and such patterns need to be learned from the data.
> >
> > I am convinced that their model is learning some higher-level patterns. However, I would be more convinced that those patterns are necessary if they showed that their model results in more valid designs than the simple baselines I propose in my review.
> >
> > 3.  I am satisfied with this, but would like to see the proposed revisions.
> > 4. If the scores are bimodal, that's yet another argument that reporting a mean and standard deviation in the main table is not the most effective way to communicate the relative performance of their model and baselines.
> > 5. Thank you for the explanation. I would like to see a clarification in the revision as well.
> > 6. To be clear, I was not suggesting that the authors try to predict structures for TCRs! Perhaps "scaffold embedding" would be clearer? o
> > 7. Wonderful!
> > 8. Thank you for the explanation. I would like to see a clarification in the revision as well.
> > 9. Thank you for the explanation. I would like to see a clarification in the revision as well.
> > 10. Thank you for satisfying my curiosity!
> >
> > With the exception of point 2, I am satisfied with the rebuttals pending a look at the actual revised paper, which I don't think has been uploaded. I know the discussion period is not that long, but I think having stronger evidence that a manual split between scaffold and function does not work as well would greatly strengthen the paper.

---

> > > ### Author Response · Authors · 2023-08-11
> > >
> > > Thanks for your replies. Due to space limit, we cannot include our revisions in the original rebuttal (and **we cannot submit a new manuscript at this stage**), but we will list our proposed revisions according to your comments here:
> > >
> > > The highlighted parts are modifications. The others are texts that will be incorporated into the manuscript.
> > >
> > > 1\. *Title*: Disentangled Wasserstein Autoencoder for **T-Cell Receptor Engineering**
> > >
> > > *Introduction*: We focus on the CDR3beta region of TCRs where there is sufficient data and a clearly defined functional role (peptide binding).
> > >
> > > *Introduction*:  To our knowledge, we are the first to formulate computational **TCR** design as a style transfer problem and leverage disentangled embeddings for **the sequence engineering**, resulting in more interpretable and efficient conditional generation and property manipulation.
> > >
> > > 2\. Due to space limits, it will come up in the next comment.
> > >
> > > 3\. *Introduction*: A related application is the motif scaffolding problem [*citations*] where a structural "scaffold" is generated supporting a fixed functional ``motif". Here our goal is the opposite: to modify the functional parts of the sequence instead by directly introducing mutations onto it.
> > >
> > > *Related work (Computational biological sequence design)*: It is also possible to co-design sequences along with the 3D structure, such as in motif scaffolding [*citations*].
> > >
> > > References:
> > > [1] Trippe, Brian L., et al. "Diffusion probabilistic modeling of protein backbones in 3d for the motif-scaffolding problem." arXiv preprint arXiv:2206.04119 (2022).
> > > [2] Wang, Jue, et al. "Scaffolding protein functional sites using deep learning." Science 377.6604 (2022): 387-394.
> > > [3] Watson, Joseph L., et al. "Broadly applicable and accurate protein design by integrating structure prediction networks and diffusion generative models." BioRxiv (2022): 2022-12.
> > >
> > > 4\. Thanks for the suggestion. We will remove the binding score column as it’s not very informative, and change it to the percentage of positive sequences. We’ll also highlight the main metric for comparison (last column), similar to the table in the response to question 2.
> > >
> > > 5\. *2.3 Disentanglement Guarantee*: Thus, our objective can jointly minimize $I(Z_f, Z_s)$ and maximize $I(X, Z_s)$ and $I(Y, Z_f)$. The former ensures independence between the embeddings, and the latter enforces them to learn separate information, achieving disentanglement.
> > >
> > >
> > > 8\. *Conclusion*: Also, our model assumes some “natural", yet unknown, disentanglement between features within the data. Thus, it could potentially be applied to other protein design tasks if there are well-defined functional terms and sufficient training data.
> > >
> > >
> > > 9\. *3.2.4 Analysis of Engineered TCRs*: We can also observe some, though less frequent, changes in the conservative regions through modifying the $z_f$, while some residues in the hypervariable region are maintained. Furthermore, the attention patterns of the functional and structural encoders, despite having their own preferences, overlap at some positions and do not have a clear-cut separation. Neither pattern align with the hypervariable/conservative separation either. These results indicate that the embeddings learn to encode patterns from the entire sequence, and do not fit into the manual separation of a “functional" hypervariable region and a “structural" conservative region.

---

> > > > ### Author Response · Authors · 2023-08-11
> > > >
> > > > 2\. We have experimented with the two baselines you suggest:
> > > >
> > > > (1) Only model the hypervariable region and use a constant consensus sequence for the first 4 and last 3 residues
> > > >
> > > > (2) Separately embed the hypervariable and conserved regions directly, instead of having the model learn them from the data.
> > > >
> > > > We train a model only using the hypervariable region and run the sequence modification pipeline as in full CDR3beta. After the hypervariable region is modified, we add back the conservative regions from two different sources listed below:
> > > >
> > > > *trimmed/ constant*: we add CASS and QYF to all generated sequences, as suggested in baseline (1).
> > > >
> > > > *trimmed/ template*: we put back the conservative region of the input template for each generated sequence. We think this is an easier way to achieve baseline (2). Since we’re not modifying the conservative region, it would be more efficient to simply put it back, instead of embedding it.
> > > >
> > > > The results are as follows:
> > > > |                    | $\bar{r_v}$   | %positive   | %valid   | **%positive valid**   |
> > > > |:-------------------|:------------|:---------------------|:----------------------|:--------------------------|
> > > > | TCR-dWAE  | 1.36±0.03   | 0.39±0.05            | 0.61±0.06          | **0.23±0.02**                 |
> > > > | TCR-dWAE (trimmed/ constant) | 1.48±0.02   | 0.21±0.06            | 0.75±0.02      |       **0.16±0.05**                 |
> > > > | TCR-dWAE (trimmed/ template) | 1.35±0.08   | 0.19±0.07            | 0.68±0.08             | **0.11±0.02**                 |
> > > >
> > > > As expected, the “trimmed” performance is lower than modeling the full sequence. To our surprise, it seems that the lower performance is caused by a massive drop in the number of positive sequences, while the sequence validity is not affected much by the trimming, as can be seen from the number of valid sequences.
> > > >
> > > > The attention maps of the model trained on trimmed data are generally similar to the middle segment (i.e. on the hypervariable region) of Attached Fig 1.
> > > >
> > > > Based on these results, we would like to qualify our assumption in the main rebuttal:
> > > >
> > > > (1) It is actually the **binding score** that is more affected by the conservative region, rather than validity. Thus, the conservative region should have some important functional information that are not yet known, and there may be interactions between the two regions to ensure positive binding.
> > > >
> > > > (2) The hypervariable region itself can still be further separated into its own “functional” and
> > > > “structural” patterns. Its structural pattern (which preserves the hypervariable region's own validity) is somewhat independent of the conservative region. Its functional pattern, however, needs to interact with that of the conservative region to make the full CDR3beta sequence positive, so it cannot be confined to the hypervariable region.
> > > >
> > > > In other words, we could consider the information content of the embeddings trained on the full CDR3beta as follows:
> > > >
> > > > 1. $z_f$: binding-related patterns that span the whole sequence.
> > > >
> > > > 2. $z_s$: patterns that maintain the validity of the hypervariable region and patterns that maintain the validity of the conservative region, which are somewhat less dependent on each other.
> > > >
> > > > We would also like to acknowledge that our analysis could be biased by the data. The point of the paper is to demonstrate a computational approach on a simple and well-defined platform. Thus, we are careful not to make too strong claims about the actual TCR structural biology beyond our setup.
> > > >
> > > > Overall, these results do not contradict with the claim that the patterns cannot be manually separated in our study. We’ll add them to the Appendix.

---

> > > > > ### Comment · Reviewer_TMVT · 2023-08-17
> > > > > **Score updated**
> > > > >
> > > > > Thank you for thoroughly addressing concerns -- I know they required a significant amount of work, but I do believe the final paper will be much improved. As a result, I have raised my score from 4 to 7.

---

### Official Review · Reviewer_mx2A · 2023-07-01

**Soundness:** 2 fair
**Presentation:** 3 good
**Contribution:** 3 good
**Rating:** 7
**Confidence:** 3

**Summary:**

In this work, the authors develop an approach to T-cell receptor (TCR) generation using disentangled representation learning (DRL). They introduce a disentangled Wasserstein autoencoder with an auxiliary classifier, and show through a series of experiments that they can separate the “structural” and “functional” representations of TCR sequences. Sample validity and quality are evaluated using pre-trained autoencoders and binding classifiers. Overall, the paper addresses an interesting and challenging problem area (TCR engineering) and DRL is a promising avenue for protein engineering more broadly. However, some questions about the evaluation procedure lead to difficulties in interpreting the results.

**Strengths:**

The paper considers an important application area, T-cell receptor engineering, which is understudied in the ML community, and is an original application for discrete sequence modeling. The overall methodology and presentation is clear.

**Weaknesses:**

Because the authors introduce TCR validity as one of their main evaluation criteria, it requires substantial evidence that it is a meaningful optimization target. Some of the details related to evaluation and limitations are not clear.

**Questions:**

Are there examples in the protein engineering literature for using a pre-trained autoencoder to calculate validity? Is the VDJDB used for training TCR-dWAE a subset of TCRdb? Have the authors considered a simpler metric like “naturalness” (pseudo log-likelihood) from a masked language model or log-likelihoods from an autoregressive language model trained on TCRdb? In the discussion of TCR validity, what does “has a similar embedding pattern as the known TCRs” mean?

The authors claim that the structural embedding constrains the sequence backbone to ensure validity, but reported %validity is < 60% for TCR-dWAE. The % positive valid is also very low, despite the low number of mutations. As the authors mention, a low number of mutations is desirable for real protein engineering tasks to improve function, but it may be unfeasible to introduce function (peptide-specific binding) with a single mutation.

In Table 2, is it the case that the average number of mutations is about 0.5 for each method? How is this reconciled with the reported uniqueness and validity?

The authors limit modeling to the CDR3beta region, which is reasonable for modeling function, but weakens any demonstrations of “disentangled” representations. There are highly conserved motifs at the beginning and end of CDR3beta that the generated samples correctly recover. It is difficult to evaluate claims of structure vs function disentanglement in the high-entropy region of CDR3beta. This argument may be better supported by showing guided sampling that interpolates between two known modes of TCR binders.

How are Miyazawa-Jernigan energies calculated?


**Limitations:**

Yes

---

> ### Author Rebuttal · Authors · 2023-08-10
>
> Thanks for your comments. Here we respond to your points and questions in the sequential order:
>
> **1. Validity metrics**
>
> We follow the same evaluation metrics as in [1]. Specifically, with a trained autoencoder, we use (1) reconstruction accuracy (2) the likelihood in the distribution of the embeddings of known TCR sequences (estimated by a Gaussian mixture model) as a metric of the “likelihood” of a new TCR. This is similar to autoencoder-based out-of-distribution (OOD) detection methods (refs below).
>
> TCRdb is a much larger database with 277 million TCRs. Most unique CDR3beta sequences in VDJDB should appear in it (we didn’t do a comprehensive search, but the majority of a subset we have tried can be found in TCRdb). However, as it is thousands of times larger than the entire VDJDB, we believe the autoencoder trained on it should learn more generic patterns that can be used for the validation.
>
> “Has a similar embedding pattern as the known TCRs” means when the generated sequences are fed to the autoencoder trained on TCRdb, their embedding patterns have a high likelihood in the distribution of the embeddings of known TCRs. We also show in Appendix Fig. S1 that our metric could separate real TCRs from other protein segments or random sequences very well. Thus, we believe it is sufficient for the evaluation.
>
> Ref:
> [1] Chen, Ziqi, et al. "T-Cell Receptor Optimization with Reinforcement Learning and Mutation Polices for Precision Immunotherapy." International Conference on Research in Computational Molecular Biology. Cham: Springer Nature Switzerland, 2023.
> [2] Pimentel, Marco AF, et al. "A review of novelty detection." Signal processing 99 (2014): 215-249.
> [3] Zong, Bo, et al. "Deep autoencoding gaussian mixture model for unsupervised anomaly detection." International conference on learning representations. 2018.
>
> **2. Number of mutations**
>
> Sorry for the confusion. In fact, the average number of mutations reported in Table 1 is divided by the sequence length, not the raw number. This means we have on average ~50% of the original sequence modified. We will clarify this in the future. Thanks for pointing this out.
>
> **3. Structural vs function within the hypervariable region; interpolation**
>
> We have done interpolation on 100 positive and negative pairs (Attached Fig E). Overall, interpolation between positive $z_f$’s generates far more positive samples than negative ones. We also observe in an example (Attached Fig B) that throughout the interpolation, changing $z_f$ only affects some parts of the hypervariable region, and sometimes it also modifies the conservative region, while some others from both regions are preserved.
>
> From these observations, we can have two conclusions:
> (1) It’s not that $z_s$ is solely responsible for the reconstruction of the conservative region.
> (2) There is still a functional/structural separation within the hypervariable region, where some positions can be modified to alter functions while some cannot.
>
> This is further supported by Attached Fig A that the average attention patterns of the functional and structural encoders, despite having their own positional preferences, overlap at some positions in the hypervariable region. Neither patterns align with the boundary of hypervariable/conservative regions.
>
> Thus, the functional and structural embeddings may have learned higher-level interactions between sequential patterns all across the sequence, including but not limited to the hypervariable region.
>
> **4. Miyazawa-Jernigan energy**
>
> Miyazawa-Jerningan energy is based on a heuristic score matrix of pairwise interaction energy between amino acids. We sum up the score of each pair of “interacting” amino acids (whose distance falls below a certain threshold; in this case, we consider all amino acids in the CDR3beta and the peptide interact with each other).

---

> > ### Comment · Reviewer_mx2A · 2023-08-11
> >
> > I have read the rebuttal and am satisfied with the response. Thank you to the authors for addressing the points raised in my review.

---

> > > ### Author Response · Authors · 2023-08-13
> > >
> > > Thanks for your reply. We would like to ask if there’s any other point we need to address so that the score of our paper can be improved. We would be happy to consider your suggestions.

---

> > > > ### Comment · Reviewer_mx2A · 2023-08-18
> > > >
> > > > I have raised my score to 7, thank you for your responses and work during the rebuttal period.

---

### Official Review · Reviewer_ULRU · 2023-07-05

**Soundness:** 4 excellent
**Presentation:** 4 excellent
**Contribution:** 3 good
**Rating:** 6
**Confidence:** 4

**Summary:**

This paper proposes to use disentangled representations for protein engineering which enables preserving the generic structure while making functional modifications to the protein of interest. To disentangle functional patterns from the rest, they propose to use disentangled Wasserstein auto encoders with auxiliary classifier on the embeddings that encode function. The application of their method has been studied for T-Cell receptor engineering. It is demonstrated that given negative pairs of peptide (TCR target) and TCR, their method can alter TCRs into new sequences with higher validity and binding score relative to other approaches.

**Strengths:**

The problem is well motivated, and the paper does a good job in adapting the approaches developed in other domains and applying it to protein engineering.

**Weaknesses:**

The proposed method is not novel but its application to a new domain is of interest.

**Questions:**

Q1. The results are averaged over four selected peptides (AUROC > 0.8) in Table 1. In the experiments section, it is stated that 10 peptides were used.

1.1.	Can you provide the results of Table 1 for the rest of the peptides?

1.2.	Also, as only 10 peptides were used, can you provide the statistics in Table 1, e.g., r_v
and r_b, for each peptide?

1.3.	Given that you engineered 5000 TCRs per peptide, I would like to see the correlation between the estimated validity and the binding score for each peptide.

Q2. Theoretically you have shown that the two latent embeddings of structure and function are encouraged to be disentangled. However, in the experiments do you have a way of measuring structural similarity between the altered TCR and the input TCR that does not bind to the peptide? I understand that the structural embedding is intact in the generation, but can you quantify it in the sequence space? I do not think that the validity score tells us whether the output folds like the input. Please clarify this.

To explain my point better, are all TCRs roughly similar in structure? Or their structural properties, e.g., folding, varies a lot. If the former is true, then maybe validity score is a good criterion but if the latter holds then you should quantify the similarity of the generated sequence with improved binding to the input sequence with poor binding.

---

> ### Author Rebuttal · Authors · 2023-08-10
>
> Thanks for your comments. Here we respond to your points and questions in the sequential order:
>
> **1. Additional results for Table 1**
>
> The other six peptides were discarded because of poor classification (with AUC < 0.8). As we rely on the classifier to control the information content of the functional embedding, poor classification leads to uninformative disentanglement, i.e. the functional embedding does not really encode the correct binding information.
>
> Statistics averaged over all 10 peptides:
> |                   | $\bar{r_v}$   | $\bar{r_b}$   |  %valid   | #mut/len   | %positive valid   |
> |:------------------|:------------|:----------|:-----------------|:--------------|:--------------------------|
> | TCR-dWAE (best)   | 1.43±0.02   | 0.37±0.09 | 0.55±0.09        | 0.5±0.05      | 0.15±0.04                 |
> | TCR-dWAE (avg pos)    | 1.48±0.07   | 0.32±0.11 | 0.66±0.11        | 0.43±0.08     | 0.15±0.03                 |
> | TCR-dWAE (random pos) | 1.47±0.02   | 0.31±0.04 | 0.76±0.03        | 0.47±0.06     | 0.18±0.01                 |
> | IDEL  (random pos)  | 1.51±0.01   | 0.26±0.03 | 0.55±0.02        | 0.38±0.02     | 0.1±0.01                  |
> | TCR-dWAE (null)   | 1.51±0.01   | 0.08±0.02 | 0.86±0.01        | 0.45±0.06     | 0.07±0.02                 |
> | original          | 1.59    | 0.03  | 0.92        | 0.0      | 0.03                  |
>
>
>
> Statistics separated by peptides, under "random pos" mode (due to space limits, we only show 2 well classified and 2 poorly classified examples, but the pattern is similar for all. The full table will be added to the new version):
>
> *FRDYVDRFYKTLRAEQASQE (AUC = 0.88)*
> |                   | $\bar{r_v}$   | $\bar{r_b}$   |  %valid   | #mut/len   | %positive valid   |
> |:----------------|:------------|:----------|:-----------------|:--------------|:--------------------------|
> | TCR-dWAE   (random pos)   | 1.59±0.03   | 0.28±0.06 | 0.86±0.03        | 0.47±0.07     | 0.22±0.04                 |
> | IDEL  (random pos)    | 1.58±0.01   | 0.35±0.05 | 0.5±0.03         | 0.39±0.02     | 0.12±0.02                 |
> | TCR-dWAE (null) | 1.51±0.01   | 0.05±0.04 | 0.86±0.01        | 0.45±0.07     | 0.04±0.03                 |
>
>
> *CTPYDINQM (AUC = 0.91)*
> |                   | $\bar{r_v}$   | $\bar{r_b}$   |  %valid   | #mut/len   | %positive valid   |
> |:----------------|:------------|:----------|:-----------------|:--------------|:--------------------------|
> | TCR-dWAE  (random pos)  | 1.36±0.03   | 0.51±0.07 | 0.66±0.04        | 0.49±0.05     | 0.29±0.04                 |
> | IDEL  (random pos)  | 1.4±0.02    | 0.37±0.07 | 0.49±0.05        | 0.44±0.02     | 0.13±0.01                 |
> | TCR-dWAE (null) | 1.5±0.01    | 0.08±0.02 | 0.85±0.01        | 0.45±0.07     | 0.06±0.02                 |
>
> *ELAGIGILTV (AUC=0.66)*
> |                   | $\bar{r_v}$   | $\bar{r_b}$   |  %valid   | #mut/len   | %positive valid   |
> |:----------------|:------------|:----------|:-----------------|:--------------|:--------------------------|
> | TCR-dWAE  (random pos)  | 1.57±0.03   | 0.1±0.01  | 0.9±0.02         | 0.44±0.06     | 0.09±0.01                 |
> | IDEL  (random pos) | 1.58±0.02   | 0.13±0.01 | 0.64±0.02        | 0.32±0.02     | 0.07±0.01                 |
> | TCR-dWAE (null) | 1.51±0.02   | 0.08±0.0  | 0.86±0.02        | 0.45±0.05     | 0.07±0.0                  |
>
> *AVFDRKSDAK (AUC = 0.64)*
> |                   | $\bar{r_v}$   | $\bar{r_b}$   |  %valid   | #mut/len   | %positive valid   |
> |:----------------|:------------|:----------|:-----------------|:--------------|:--------------------------|
> | TCR-dWAE   (random pos)    | 1.58±0.03   | 0.15±0.01 | 0.9±0.02         | 0.45±0.06     | 0.13±0.01                 |
> | IDEL   (random pos)    | 1.59±0.02   | 0.19±0.02 | 0.64±0.03        | 0.34±0.02     | 0.1±0.01                  |
> | TCR-dWAE (null) | 1.51±0.01   | 0.15±0.01 | 0.85±0.01        | 0.46±0.06     | 0.12±0.01                 |
>
> We also exclude some baselines as they are uniformly poor.
>
> For the poorly classified peptides (AUC < 0.8),  using a positive $z_f$’s is not much different from using a random $z_f$ (the "null" row), meaning in this case the $z_f$ is uninformative.
>
> The correlation between $r_v$ and $r_b$ by peptide is shown in Attached Fig. C. As expected, there is no significant correlation.
>
> **2. Structural similarity**
>
> We are focusing on the CDR3beta region, a loop whose structure is difficult to predict. We do acknowledge that it’s hard to directly quantify structural similarities with our current method, due to lack of 3D structural data and reliable prediction methods. Thus, our “structural pattern” can only be used to indirectly measure structural similarity through sequence similarity. Specifically, the autoencoder for the validation could learn frequent sequential patterns and their importance among the a set of known TCRs, and we assume that this indirectly implies structural similarities, based on the assumption that the structure is defined by the sequence. We will state these limitations in the next version.

---

> > ### Comment · Reviewer_ULRU · 2023-08-12
> >
> > Thanks. All my questions have been addressed. I will maintain my original score.

---

> > > ### Author Response · Authors · 2023-08-13
> > >
> > > Thanks for your reply. We would like to ask if there’s any other point we need to address so that the score of our paper can be improved. We would be happy to consider your suggestions.

---

### Author Rebuttal · Authors · 2023-08-10

We really appreciate the reviewers’ valuable comments. Here we would like to address some common concerns that several reviewers have brought up.

We have also attached a pdf with the figures for the rebuttal, and refer to them as “Attached Fig XX”.

**1. The usage of “structure embedding”**

We agree that the wording is a little confusing. The “structure” here means “sequential patterns that relate to the 3D structural backbone”.  For simplicity, we use the word “structural” in the rest of the paper. However, we are also open to suggestions of more accurate wording.

In practice, it is difficult to predict the real “structure” in the sense of protein 3D structure for CDR3beta region because it is a very flexible loop, and high-quality structure is scarce. Additionally, it cannot be predicted from the CDR3beta sequence alone, ignoring the rest of the TCR. Thus, we can only rely on the available sequential information of known CDR3beta's to determine whether generated ones are valid, or whether they preserve the structural backbone, based on the intuition that the structure is defined by the sequence. Thus,  if certain pivotal residues and motifs (which we try to make the “structural embedding” learn) are similar between two sequences (e.g. generated and known), they should have similar structures.

**2. Separating structures and functions**

There are also concerns that the separation of “structural” and “functional” embedding in the latent space is unnecessary, as CDR3beta already has a conservative region (first four and last three residues) and a hypervariable region, which can be manually defined as the “structural” and “functional” parts of the sequence. Moreover, it might be possible to remove the conservative region entirely as the structural embedding might only learn to memorize those residues.

We would like to address that from the following perspectives:

(1) There are actually quite a few variations within the conservative regions in real TCRs. It is uncertain whether it interacts with the hypervariable region to maintain validity. For example, some combinations might be considered invalid. Thus, we think it is necessary to still include it into the modeling.

(2) From the engineering experiments, we discover that by only modifying $z_f$, there are changes in both the conservative region and the hypervariable region, though the latter is more frequent. On the other hand, the variable region is not fully replaced and some residues are maintained. This can be seen in the mutation hotspots in Fig. 2B and the interpolation example in Attached Fig. B. For the interpolation experiment, we obtain the embedding of two sequences (CASTESDRRSQNTQYF and  CASSLSTFTANTAQLFF) that binds to peptide CTPYDINQM. We interpolate their  $z_f$ by  $z^{(r)}_f = rz^{(1)}_f + (1-r)z^{(2)}_f, r\in[0,1]$, and combine it with the $z_s$ of each sample to do the generation. We observe that throughout the process, some amino acids in both regions are gradually modified from one binding mode to another while the rest remains.

(3) We further show in the Attached Fig. A that the attention patterns of the functional and structural encoders, despite having their own positional preferences, also overlap at some positions. In addition, neither agrees with the boundary of the conservative region and the hypervariable region.

These observations indicate that both the structural and functional patterns the model learns are higher-level patterns and could span both the conservative and hypervariable regions, and cannot be fit into manually defined windows. Thus, we believe a manual separation or exclusion of certain residues is not possible, and such patterns need to be learned from the data.


**3. Generalization to other proteins**

Deep learning-based protein engineering is limited by the availability of data. In our specific scenario, we need both a well-defined functional label, and samples that are large enough for the machine learning. To our knowledge, there have not been a lot of problems that meet both requirements. Because of these concerns, we limit our experiments to the CDR3beta region of TCR.

Nevertheless, we do agree that “protein engineering” is too broad a claim for the experiments in the paper. Thus, we will change the title to TCR engineering, and clearly state the limitations of our study in the next version.

**4. Updated Table 1**

In response to some reviewers’ suggestions, we have updated the results in Table 1 in two ways: (1) we have included the results of 1 additional random seed, resulting in 5 in total; (2) as IDEL is a VAE, we do random generation 5 times given the mean and variance of $z_f$ and $z_s$ and choose the one with the highest binding score, instead of directly using the mean as in the previous Table 1. “Best” and “avg” modes do not apply to this generation procedure so we exclude them for IDEL. Some baselines are not included here because their performance remains low after the change.

|                   | $\bar{r_v}$   | $\bar{r_b}$   |  %valid   | #mut/len   | %positive valid   |
|:------------------|:------------|:----------|:-----------------|:--------------|:--------------------------|
| TCR-dWAE (best)   | 1.33±0.07   | 0.53±0.18 | 0.45±0.14        | 0.5±0.05      | 0.18±0.05                 |
| TCR-dWAE (avg pos)    | 1.37±0.13   | 0.51±0.21 | 0.53±0.18        | 0.46±0.09     | 0.2±0.06                  |
| TCR-dWAE (random pos) | 1.36±0.03   | 0.48±0.09 | 0.61±0.06        | 0.49±0.05     | 0.23±0.02                 |
| IDEL  (random pos)  | 1.44±0.02   | 0.35±0.05 | 0.45±0.04        | 0.42±0.01     | 0.1±0.01                  |
| TCR-dWAE (null)   | 1.51±0.01   | 0.05±0.03 | 0.85±0.01        | 0.45±0.07     | 0.04±0.03                 |
| original          | 1.59    | 0.01  | 0.92         | 0.0       | 0.01                  |

---

> ### Comment · Reviewer_TMVT · 2023-08-10
> **Acknowledgement + pdf doesn't work**
>
> I've read the rebuttals and will respond soon in more detail. However, I cannot get the attachment to load.

---

> > ### Author Response · Authors · 2023-08-10
> >
> > Thanks for your reply. We are preparing a response to it and will post it later.
> >
> > For the attachment, we can open it on our side. One of the figures has many data points so it might take a while to load.

---

### Decision · Program_Chairs · 2023-09-21

**Decision:**

Accept (poster)

**Comment:**

This paper proposes a method for T-cell receptor engineering, which is designed to preserve the 3D structure while making functional changes (i.e. binding) to the protein of interest. The method consists of training a disentangled Wasserstein autoencoder with an auxiliary classifier to learn separate functional and structural embeddings. The reviewers all agreed that the proposed method (disentangled WAE) is not novel but its application to a new and important domain is interesting and well-designed. Reviewers were somewhat skeptical of the evaluation and potential limited applicability of this approach, however, the authors addressed these concerns during the rebuttal period and will incorporate their feedback into the manuscript, including updating the title.